# THE EFFECT OF DEPTH ON THE EXPRESSIVITY OF DEEP LINEAR STATE-SPACE MODELS

## ABSTRACT

Deep state-space models (SSMs) have gained increasing popularity in sequence modelling. While there are numerous theoretical investigations of shallow SSMs, how the depth of the SSM affects its expressiveness remains a crucial problem. In this paper, we systematically investigate the role of depth and width in deep linear SSMs, aiming to characterize how they influence the expressive capacity of the architecture. First, we rigorously prove that in the absence of parameter constraints, increasing depth and increasing width are generally equivalent, provided that the parameter count remains within the same order of magnitude. However, under the assumption that the parameter norms are constrained, the effects of depth and width differ significantly. We show that a shallow linear SSM with large parameter norms can be represented by a deep linear SSM with smaller norms using a constructive method. In particular, this demonstrates that deep SSMs are more capable of representing targets with large norms than shallow SSMs under norm constraints. Finally, we derive upper bounds on the minimal depth required for a deep linear SSM to represent a given shallow linear SSM under constrained parameter norms. We also validate our theoretical results with numerical experiments.

## 1    INTRODUCTION

Recent advances in state space models (SSMs) have been successful in learning long sequence relationships via mitigating the computational inefficiency of explicitly modeling token interactions (Gu & Dao, 2024; Gu et al., 2022; 2020; 2021; Smith et al., 2023). It achieves significantly better performance compared with attention-based transformers in the long range arena (LRA) dataset (Tay et al., 2021). The linear time-invariant structure of SSM allows for an asymptotic computational complexity of only $O(T \log T)$, which is significantly better than the $O(T^2)$ complexity of traditional full-attention approaches exhibiting significant computational demands (Vaswani et al., 2017). Moreover, SSMs have proven effective in multiple domains dealing with continuous signal data, including audio and vision tasks (Li et al., 2024; Goel et al., 2022; Nguyen et al., 2022).

Despite its success across various fields of practical application, many theoretical questions remain unanswered. One of the biggest problems is the expressivity of deep state-space models. Although modern SSMs (Smith et al., 2023; Gu & Dao, 2024) comprise dozens of layers and contain millions of parameters, it is still unclear what the real difference is between shallow SSMs and deep SSMs, and the reason why we need depth in state-space models. Another notable problem is what kind of sequence to sequence relationship can be handled better with a deep network than with a shallow network.

A number of recent works have employed dynamical systems approaches to study token stability in deep SSMs (Vo et al., 2025). Geshkovski et al. (2023) uses a similar dynamics-based formulation to investigate feature clustering behaviors in transformers. Other studies focus on the training dynamics of deep linear networks from an optimization perspective (Menon, 2024; Du & Hu, 2019), whereas we focus on representation with norm constraints. Especially, Du & Hu (2019) analyzes the benefits of width in deep linear networks, showing that deep and wide linear neural networks converge to a global minimum with polynomial running time while deep and narrow linear networks converge with exponential running time. In this paper, we consider deep linear state-space models (SSMs) which apply to sequence modeling. This fundamental difference leads to distinct findings in investigating the two models. However, relatively little work has been devoted to understanding how depth affects the

expressivity of deep SSMs. We adopt a simple formulation that allows us to characterize important differences between multi-layer SSM and one-layer SSM. Our main contributions are presented as follows:

- In Theorem 1, we prove that, in the absence of norm constraints, depth and width are equivalent in approximation in the sense that under a fixed parameter budget, models of arbitrary depth achieve the same expressive power.

- In Theorem 2, we demonstrate that a shallow SSM with large-norm weights can be exactly represented by a deep SSM with smaller-norm weights, under explicit norm constraints on the model parameters.

- We establish upper bounds on the minimal depth required for a deep linear SSM to represent a given shallow linear SSM in the regime of constrained parameter norms in Theorem 3 and validate our theoretical results with numerical experiments.

## 2 RELATED WORKS

**Expressivity of State-Space Models**   State-space models originate from the HIPPO matrix, which is optimal in the online function approximation sense (Smith et al., 2023; Gu et al., 2020). Wang & Xue (2023) also provides theoretical guarantees for the approximation of continuous sequence-to-sequence mappings using SSMs with layer-wise nonlinear activations. Furthermore, Muca Cirone et al. (2024) employed tools from Rough Path Theory to show that deep diagonal SSMs possess less expressive power than their non-diagonal counterparts. However, these works primarily focus on function approximation of deep SSMs, whereas the present work aims to present the simplest setting where one can study how depth influences the expressivity in deep SSMs.

**Dynamics in deep State-Space Models**   Several works focus on dynamics in deep SSMs. Vo et al. (2025) investigates the divergence behavior of tokens in a pre-trained Mamba model by characterizing continuous-time systems. Smekal et al. (2024) shows how the memory of a deep linear SSM varies with the depth and width, and how the learning dynamics of deep linear models vary with memory expressivity. Our work focuses on a different setting and complements existing analyses by providing a theoretical understanding of depth in deep linear SSMs.

**Deep linear networks**   Simple linear architectures often serve as effective tools for gaining theoretical insights into the behavior of deep neural networks. Arora et al. (2019) proves that the convergence of gradient descent achieves a linear rate for training a deep linear neural network over whitened data under certain conditions. Bah et al. (2022) also shows that optimizing a deep linear network is equivalent to Riemannian gradient flow on a manifold of low-rank matrices, with a suitable Riemannian metric. Gruber & Avron (2024) study the implicit bias arising from weight initialization in deep linear networks, offering the insight that weight norms are central to understanding the behavior of deep models. Following prior theoretical analysis of deep linear networks, our work focuses on understanding the expressivity of deep recurrent models, specifically deep state-space models.

## 3 PROBLEM FORMULATION

One particularly notable feature of currently popular sequential models such as S4 (Smith et al., 2023) and Mamba (Gu & Dao, 2024) is the stacking of multiple layers, which leverages scaling laws to increase the number of parameters and achieve stronger performance. In this section, we aim to provide a theoretical perspective on the architecture of deep state-space models, to characterize more precisely how depth contributes to their success.

## 3.1 DEEP LINEAR STATE-SPACE MODELS

For a general $l$-layer deep state-space model, we express its recurrent form as follows, where each layer's input comes from the previous layer's output:

$$
\begin{aligned}
y(t) &= C^T \sigma(h_l(t)) \\
h_l(t+1) &= A_l h_l(t) + B_l \sigma(h_{l-1}(t+1)) \\
&\vdots \\
h_2(t+1) &= A_2 h_2(t) + B_2 \sigma(h_1(t+1)) \\
h_1(t+1) &= A_1 h_1(t) + B_1 x(t+1)
\end{aligned}
\tag{1}
$$

where $h_l(t) \in \mathbb{R}^m$ is the hidden state of $l$-th layer at time step $t \in \mathbb{N}$. $\sigma : \mathbb{R}^{m \times 1} \to \mathbb{R}^{m \times 1}$ is activation function on hidden state $h_l(t)$. $A_1, ..., A_l, B_2, ..., B_l \in \mathbb{C}^{m \times m}$ and $C, B_1 \in \mathbb{C}^{m \times 1}$ are hidden matrices. Also, $A_1, ..., A_l$ are the state-space matrices, $B_1, ..., B_l$ are the input matrices and $C$ is the output matrix. We assume the standard zero-state initialization for SSMs, i.e., $h_{\tilde{l}}(-1) = 0$, $\forall \tilde{l} = 1, ..., l$. For each scalar input $x(t) \in \mathbb{R}$, we obtain an output $y(t) \in \mathbb{R}$ by passing it through this $l$-layer model.

Now, we set $\sigma$ to be the identity mapping and consider a linear setting. We observe that both shallow and deep state-space models share a convolutional structure, differing only in their kernels when the nonlinearity in the function $\sigma$ is removed, which serves as a natural and tractable starting point for investigating the role of depth in SSMs.

The convolutional kernel allows us to clearly understand how the input is mapped to the corresponding output through the sequential model, i.e. $y(t) = (\rho * x)(t)$. We characterize the convolutional kernel of the deep state-space model as follows:

**Lemma 1.** *For $l$ layer deep linear SSM defined in Equation (1), the convolutional kernel $\rho(t)$ admits the following representation*

$$
\rho(t) = \sum_{\substack{i_1 + i_2 + ... + i_l = t \\ i_1, ..., i_l \in \mathbb{N}}} C^T \prod_{j=1}^{l} (A_{l-j+1}^{i_{l-j+1}} B_{l-j+1})
\tag{2}
$$

A proof of Lemma 1 can be found in Appendix A.1. Our approach to proving this result is checking the basic case and then performing induction on both timesteps $t$ and layers $l$.

Notably, it is the simplified linear architecture of our network that enables the convolutional kernel to be expressed in a clear and tractable form, as the sum over layer indices of successive powers of the hidden matrices. This lemma reveals how, in deep SSMs, each input $x(t)$ is convolved through such a kernel to produce the output $y(t)$. This insight lays the groundwork for our subsequent comparison of the representational capacities of deep SSMs and shallow SSMs.

## 3.2 NORM-CONSTRAINED HYPOTHESIS SPACE FOR DEEP LINEAR SSMS

To provide a mathematical description of the function space of deep state-space models, leveraging Lemma 1, we define the following hypothesis spaces for deep linear state-space models under norm constraints.

$$
\begin{aligned}
\mathcal{H}_{c,l}^m = \{ \rho(t) : {} & y(t) = (\rho * x)(t), A_1, ..., A_l \in \mathbb{C}^{m \times m} \text{ diagonal}, B_2, ..., B_l, \in \mathbb{C}^{m \times m}, \\
& C, B_1 \in \mathbb{C}^{m \times 1}, \max_{i=1,...,l} r(A_i) < 1, ||C||_\infty \le c, ||B_1||_\infty \le c, \max_{2 \le k \le l} \max_{1 \le i,j \le m} |(B_k)_{ij}| \le c \}
\end{aligned}
\tag{3}
$$

where $r(\cdot)$ refers to spectral radius of each matrix, $|| \cdot ||$ means the infinity norm of a vector, $m$ is the width of network, $c$ is norm constraint for parameters and $l$ is the number of layers.

Here, we assume that each $A_i$ is diagonal, a structure commonly adopted in real-world models (Saon et al., 2023). We further impose that the spectral radius of each $A_i$ is strictly less than 1 to ensure system stability. Since the implementation in Gupta et al. (2022) is one-dimensional, we assume both the input $x(t)$ and output $y(t)$ are scalar-valued. The norm constraint $c$ remains an important

factor in distinguishing the expressivity of deep state-space models, for the simple reason that the convolutional kernel is directly determined by the parameters $B_i$ and $C$. Thus, altering the norm can have a significant impact on the model's expressivity. To better characterize two SSMs are equivalent, we give the following definition:

**Definition 1.** *We say that a one-layer SSM and an $l$-layer SSM are equivalent if they induce the same convolution kernel $\rho(t)$ defined in Lemma 1, i.e., they implement the same linear map $x \mapsto y$ with $y(t) = (\rho * x)(t)$.*

It is a natural thing to consider norm constraints because of optimization. Due to the choice of optimization algorithms (e.g. weight decay) or initialization schemes (small norm initialization), the parameter norm of each layer would be effectively upper bounded by some constant. State-space models may suffer from large parameter norms, which may lead to training instability and negatively impact model performance (Jiang et al., 2025; Jelassi et al., 2024). Especially, Liu & Li (2025) shows that the parameter norm of SSMs would be at least exponentially increasing with respect to width if we would like to keep the same budgets to approximate specific target relationships. Here we give an explicit characterization:

**Proposition 1.** *For one-layer linear SSM (defined in Equation (1) with $l = 1$ and we remove the nonlinearity $\sigma$), we fix a diagonal matrix $A_1 = \text{diag}(\alpha_1, \alpha_2, \ldots, \alpha_m)$ with spectral radius less than 1 and $B = (1, ..., 1)^T$. Our target is the impulse memory function defined as $\rho_k(t) = 1$ if $t = k$ and $\rho_k(t) = 0$ if $t \neq k$. For each integer $k \geq 0$ and $0 < \epsilon < 1$, we define*

$$\mathcal{C}_{m,k}(\epsilon) := \{C \in \mathbb{C}^m : \max_{t \geq 0} |\rho_C(t) - \rho_k(t)| \leq \epsilon\}$$

*where $\rho_C(t)$ is convolution kernel corresponding to one-layer SSM following Lemma 1. If for any $C \in \mathcal{C}_{m,k}(\epsilon)$, the impulse position grows linearly with width, i.e., $k \geq \beta m$ for some $\beta > 0$, then we have $||C||_\infty \geq \exp(\gamma m)$, where $\gamma > 0$ is constant depend on $\beta$ and $\epsilon$.*

A detailed proof of Proposition 1 is provided in Appendix B. For a fixed approximation budget, Proposition 1 shows that a one-layer SSM with a fixed diagonal state-transition matrix $A$ requires output weights whose norm grows at least exponentially with width to approximate impulse function with growing positions. Consequently, merely increasing the width of a single-layer SSM is fundamentally limited in expressivity, which motivates our focus on depth: deep SSMs with the same parameter budget can redistribute this large norm across layers, allowing the norms of the parameters in each layer to be much smaller.

# 4 MAIN RESULTS

In this section, we present our main results on the expressivity of deep state-space models within the framework defined in Section 3.2.

## 4.1 EQUIVALENCE OF DEPTH AND WIDTH WITHOUT NORM CONSTRAINTS

First, we focus on the hypothesis space of deep linear SSMs without norm constraints. Our goal is to characterize the fundamental differences between shallow and deep SSMs, which leads us to the following questions: given a one-layer SSM with a certain width, how wide must a $l$-layer SSM be in order to represent it? In previous work, Smekal et al. (2024) provided an example demonstrating how a four-layer linear SSM can be converted into a single-layer one. Here, we give a complete characterization.

**Theorem 1.** *Let $m, l \geq 1$. Recall that $\mathcal{H}_{\infty,l}^m$ denotes the hypothesis space of linear SSMs with $l$ layers and width $m$ without any norm constraint on the parameters. Then, we have*

$$\mathcal{H}_{\infty,1}^{l(m-1)+1} \subseteq \mathcal{H}_{\infty,l}^m \subseteq \mathcal{H}_{\infty,1}^{lm}$$
$$\mathcal{H}_{\infty,1}^{l(m-1)+2} \not\subseteq \mathcal{H}_{\infty,l}^m \tag{4}$$

A detailed proof of Theorem 1 based on explicit construction can be found in Appendix A.3.

On the one hand, Theorem 1 shows that given an $l$-layer linear SSM with width $m$, we can always construct a one-layer linear SSM with width $ml$ to represent it. On the other hand, the maximal width

of one-layer SSM that can be represented by an $l$-layer SSM with width $m$ is $l(m-1)+1$. This implies that there exists a convolutional kernel of one-layer linear SSM with width $l(m-1)+2$ that cannot be represented by the kernel of an $l$-layer SSM with width $m$. We have constructed such a kernel to illustrate the optimality of this width bound, as detailed in Appendix A.3. This result illustrates that $l$-layer SSM of width $m$ has expressivity equivalent to one-layer SSM of width $O(lm)$ under the same parameter count, highlighting that width can be traded for depth without loss of expressive power.

In fact, a more general result holds when the assumption that each hidden matrix $A_i$ is diagonal is relaxed to the case where $A_i$ is diagonalizable, which is a dense and open set in the matrix space $\mathbb{C}^{m \times m}$. See the appendix Appendix A.3 for details.

### 4.2 Non-equivalence of Depth and Width with Norm Constraints

In the absence of norm constraints, increasing width and depth are equivalent under the same parameter count (Theorem 1). However, under norm constraints, the effects of depth and width on the expressivity of deep linear SSMs differ significantly. Now, we focus on the hypothesis space defined in Section 3.2 to investigate the impact of norm constraints on the expressivity of deep linear state-space models. The following theorem demonstrates how a shallow SSM with a large parameter norm can be equivalently represented by a deeper SSM composed of layers with smaller parameter norms:

**Theorem 2.** *Suppose $m, l \geq 1$ and let $c_1 > 0$ be the norm constraint for the one-layer linear SSM. Consider any convolutional kernel $\rho \in \mathcal{H}_{c_1,1}^{l(m-1)+1}$ realized by a one-layer linear SSM of width $l(m-1)+1$ whose parameters satisfy the norm constraints $c_1$, then there exists an $l$-layer linear SSM of width $m$ which is equivalent to this one-layer SSM and its parameters' norm satisfies the following upper bound:*

$$\sup_{\rho \in \mathcal{H}_{c_1,1}^{l(m-1)+1}} \inf\{c_2, : c_2 > 0, \ \rho \in \mathcal{H}_{c_2,l}^m\} \leq 2c_1^{\frac{2}{l+1}} \tag{5}$$

A detailed proof of Theorem 2 can be found in Appendix A.3. Notably, under the same order of magnitude of parameter count, a given one-layer linear SSM with a large norm constraint $c_1$ can be equivalently represented by an $l$-layer SSM with width $m$ and a smaller norm constraint bounded by $2c_1^{\frac{2}{l+1}}$. Hence, as the number of layers increases, the corresponding norm of weights decreases very quickly, indicating that depth plays an important role in reducing the norm from the approximation perspective.

Here we give an example showing the construction of converting a known 1 layer linear SSM with width $2m - 1 = 7$ into a 2 layer linear SSM with width $m = 4$. Given distinct non-zero complex numbers $|\alpha_1| \leq \cdots \leq |\alpha_7|$, we suppose that the 1 layer SSM is defined by $B = C = (z_1, \ldots, z_7)^T$ and $A = Diag\{\alpha_1, \ldots, \alpha_7\}$. Let $c_0 = \max_{1 \leq i \leq 7} |z_i|$ and the $\rho$ defined by the above SSM satisfies $\rho \in \mathcal{H}_{c_0,1}^7$, with $\rho(t) = \sum_{i=1}^7 z_i^2 \alpha_i^t$.

Then, we consider constructing a 2 layer SSM with the same $\rho$. Let $Z_0 = 2c_0^{\frac{2}{3}}$. The following constructed $A_1, A_2, B_1, B_2, C$ defines the corresponding 2 layer SSM of width 4.

- $A_1 = Diag\{\alpha_1, \alpha_2, \alpha_3, \alpha_4\}$.

- $A_2 = Diag\{\alpha_5, \alpha_6, \alpha_7, 0\}$.

- $B_1 = C = (Z_0, Z_0, Z_0, Z_0)^T$.

- $B_2 = \begin{pmatrix} \frac{(\alpha_5 - \alpha_1)z_5^2}{\alpha_5 Z_0^2} & 0 & 0 & 0 \\ 0 & \frac{(\alpha_6 - \alpha_2)z_6^2}{\alpha_6 Z_0^2} & 0 & 0 \\ 0 & 0 & \frac{(\alpha_7 - \alpha_3)z_7^2}{\alpha_7 Z_0^2} & 0 \\ \frac{(z_1^2)+(z_5^2)\frac{\alpha_1}{\alpha_5}}{Z_0^2} & \frac{(z_2^2)+(z_6^2)\frac{\alpha_2}{\alpha_6}}{Z_0^2} & \frac{(z_3^2)+(z_7^2)\frac{\alpha_3}{\alpha_7}}{Z_0^2} & \frac{z_4^2}{Z_0^2} \end{pmatrix}$

Then by calculating the convolutional kernel $\hat{\rho}$ of the above 2-layer SSM, we have

$$\hat{\rho}(t) = z_4^2\alpha_4^t + \sum_{i=1}^{3}([(z_i^2) + (z_{i+4}^2)\frac{\alpha_i}{\alpha_{i+4}}]\alpha_i^t + \frac{(\alpha_{i+4} - \alpha_i)z_{i+4}^2}{\alpha_{i+4}}\sum_{s=0}^{t}\alpha_i^s\alpha_{i+4}^{t-s}) \tag{6}$$

$$= z_4^2\alpha_4^t + \sum_{i=1}^{3}([(z_i^2) + (z_{i+4}^2)\frac{\alpha_i}{\alpha_{i+4}}]\alpha_i^t + \frac{(\alpha_{i+4} - \alpha_i)z_{i+4}^2}{\alpha_{i+4}}\frac{\alpha_{i+4}^{t+1} - \alpha_i^{t+1}}{\alpha_{i+4} - \alpha_i}) \tag{7}$$

$$= z_4^2\alpha_4^t + \sum_{i=1}^{3}([(z_i^2) + (z_{i+4}^2)\frac{\alpha_i}{\alpha_{i+4}} - (z_{i+4}^2)\frac{\alpha_i}{\alpha_{i+4}}]\alpha_i^t + \frac{z_{i+4}^2\alpha_{i+4}^{t+1}}{\alpha_{i+4}}) \tag{8}$$

$$= z_4^2\alpha_4^t + \sum_{i=1}^{3}(z_i^2\alpha_i^t + z_{i+4}^2\alpha_{i+4}^t) \tag{9}$$

$$= \rho(t) \tag{10}$$

Then the above is an explicit construction of converting a 1 layer linear SSM with width 7 into a 2 layer linear SSM with width 4 under the same parameter count with norm constraints. Actually, in the case of converting into $l$ layer SSM, the positions of non-zero elements of $B_2, \ldots, B_l$ are the same as those $B_2$ above.

From $|\alpha_i| \leq |\alpha_{i+4}|$, all the terms on the $m$-th column of $B_2$ are bounded by $\frac{\max\{|z_i^2| + |z_{i+4}^2|\}}{Z_0^2} \leq Z_0 = 2c_0^{\frac{2}{3}}$, matching **Theorem 2**. The core idea is as follows: suppose that the 1-layer SSM have weight norm $c_0$, the $l$ layer SSM decomposes the weight into a product of $l + 1$ new weight matrices, each can take a norm of order $O(c_0^{\frac{2}{l+1}})$. A similar argument holds for the case where $c_0 < 1$, in which the norms of the weights of the deep SSM can be increased.

We also conduct numerical experiments to verify our main theorems. Specifically, given a one-layer linear SSM with width $l(m-1) + 1$, we apply the construction from Theorem 2 to reconstruct an equivalent model with $l$ layers and width $m$, where a wide and shallow network is represented using a deep linear network as defined in Equation (1). We use a one-layer deep linear state-space model (SSM) with width 2521 as our target model and use deep linear SSMs defined as Equation (1) with varying depths ranging from 2 to 63 to reconstruct the target model while controlling the same parameter counts across different depths. In particular, we observe that the maximum norms decrease at the rate predicted by Theorem 2 as the number of layers increases. More details about numerical verification of Theorem 2 could be found in Appendix C.

### 4.3 MINIMAL DEPTH FOR REPRESENTING SHALLOW NETWORKS

Following Section 4.2, we have highlighted the important role of depth in enabling norm reduction in deep linear state-space models. In this section, we address a more refined question: suppose we have a SSM with weight norms bounded by $c_1$, which is possibly a very large value, how deep must a norm-constrained deep linear SSM with norm bound $c_2$ be to achieve the same expressive capacity? In fact, very large parameter norms may arise when using SSMs to learn non-smooth or highly oscillatory memory function (Jiang et al., 2025; Liu & Li, 2025) while $c_2$ could be predetermined based on desired stability constraints imposed by the optimizer. Theorem 2 already indicates that substantial norm reduction is possible through increased depth and we now formalize this by deriving the minimal depth required for a deep linear SSM with norm bound $c_2$ to represent a given one-layer linear SSM constrained by $c_1$.

**Theorem 3.** *Suppose $\rho \in \mathcal{H}_{c_1,1}^{K+1}$ for some $K \geq 2$, and let $c_1 > 1$ and $c_2 > 2$ denote the norm constraints for the one-layer and deep linear SSM, respectively. Then, we have the following upper bound for the depth of a deep linear SSM as defined in Equation* (1)*:*

$$\min\{l : \rho \in \mathcal{H}_{c_2,l}^{\lceil\frac{K}{l}\rceil+1}\} \leq \lceil\frac{2\ln(c_1)}{\ln\left(\frac{c_2}{2}\right)} - 1\rceil \tag{11}$$

A detailed proof of Theorem 3 can be found in Appendix A.3. The technique employed to determine the minimal required depth closely follows the constructive approach developed in Theorem 2.

To maintain a constant parameter count, we set the width of an $l$-layer linear SSM to be $\lceil \frac{K}{l} \rceil + 1$. If the total number of parameters is allowed to increase, the problem becomes trivial, as parameter norms can be reduced simply by allocating more parameters. However, our results in Theorem 3 indicate that increasing the depth is an efficient strategy even under a fixed parameter budget. This suggests that model performance may be improved by increasing depth if the parameter norm is large. We will show later in our experiments Section 5.

## 4.4 BEYOND DIAGONAL CASE

Recall that Theorem 1 extends to the case where the hidden matrices are diagonalizable. However, extending Theorem 2 to diagonalizable matrices remains challenging due to the condition number induced by diagonalization. To obtain tractable norm bounds, we instead assume that each hidden matrix $A_i$ is normal, i.e., unitarily diagonalizable with a condition number 1, which resolves this issue. This assumption also aligns with the HiPPO initialization commonly used in practice (Gu et al., 2020). Now we define the following hypothesis space:

$$
\begin{aligned}
\mathcal{G}_{c,l}^m = \{ \rho(t) : \ & y(t) = (\rho * x)(t), A_1, ..., A_l \in \mathbb{C}^{m \times m} \text{ normal}, B_2, ..., B_l, \in \mathbb{C}^{m \times m}, \\
& C, B_1 \in \mathbb{C}^{m \times 1}, \max_{i=1,...,l} r(A_i) < 1, \|C\|_\infty \le c, \|B_1\|_\infty \le c, \max_{2 \le k \le l} \max_{1 \le i,j \le m} |(B_k)_{ij}| \le c \}
\end{aligned}
\tag{12}
$$

The following result generalizes Theorem 2 to the normal case.

**Corollary 1.** *Suppose $m, l \ge 1$ and let $c_1 > 0$ be the norm constraint for the one-layer linear SSM. Then, we have the following upper bound for the norm constraint of equivalent $l$-layer linear SSM, where the hidden matrix $A_i$ is normal.*

$$
\max_{\rho \in \mathcal{G}_{c_1,1}^{l(m-1)+1}} \min_{c_2 > 0} \{ c_2, \ \rho \in \mathcal{G}_{c_2,l}^m \} \le 2((l(m-1)+1)c_1^2)^{\frac{1}{l+1}}
\tag{13}
$$

A detailed proof of Corollary 1, as a generalization of Theorem 2, can be found in Appendix A.3. It is important to note that, unlike the upper bound in Theorem 2, this norm upper bound explicitly depends on both the number of layers $l$ and width $m$, which reveals the fundamental difference between deep and shallow networks in their expressive ability in a larger space. Precisely, unlike the unchanged bound in Theorem 2, the bound for this larger space is increasing at the rate of $m^{\frac{1}{l+1}}$, which can also be bounded by a constant if $l = O(\log(m))$.

## 5 EXPERIMENTS

In this section, we validate our theoretical results through numerical experiments. Before presenting our results, we provide Lemma 2 that facilitates the computation of the output coefficients of a deep linear SSM as described in Equation (1), i.e., its direct expansion into the equivalent one-layer SSM form.

**Lemma 2.** *Consider the deep linear SSM in Equation (1) with $l$ layers and width $m$ and assume that each state matrix is diagonal with pairwise distinct entries $A_i = diag(\lambda_{i1}, ..., \lambda_{im})$, $i = 1, ..., l$. Suppose $B_i = (b_{rs}^{(i)})_{r,s=1}^m$, $C^T = (c_1, \cdots, c_m)$ and $B_1 = (b_1, \cdots, b_m)$. Then the convolution kernel $\rho(t)$ of the $l$ layer deep SSM with width $m$ can be written as a sum of exponentials:*

$$
\rho(t) = \sum_{\tilde{l}=1}^l \sum_{\tilde{m}=1}^m \xi_{\tilde{l},\tilde{m}} \lambda_{\tilde{l},\tilde{m}}^t
$$

*where for each layer $\tilde{l} \in \{1, ..., l\}$ and index $\tilde{m} \in \{1, ..., m\}$, the coefficient $\xi_{\tilde{l},\tilde{m}}$ is given by*

$$
\xi_{\tilde{l},\tilde{m}} = \sum_{j_1=1}^m ... \sum_{j_{l-\tilde{l}}=1}^m \sum_{j_{l-\tilde{l}+2}=1}^m ... \sum_{j_l=1}^m \frac{c_{j_1} b_{j_1 j_2}^{(l)} b_{j_2 j_3}^{(l-1)} \cdots b_{j_{l-1} j_l}^{(2)} b_{j_l}}{\prod_{\alpha=1, \alpha \ne \tilde{l}}^l (1 - \frac{\lambda_{\alpha, j_{l-\alpha+1}}}{\lambda_{\tilde{l}, j_{l-\tilde{l}+1}}})}
\tag{14}
$$

A detailed proof can be found in Appendix A.4.

## 5.1 EXPERIMENTS ON LINEAR FUNCTIONALS

In this section, we focus on the task of learning a linear functional impulse, which is a common experiment target to test the memory of sequential models (Jiang et al., 2025). Similar to the copying task (Jelassi et al., 2024), where output is a shift of the input, learning this target requires large parameter norms (Liu & Li, 2025). Thus, this task is particularly well-suited for highlighting the role of depth in deep linear state-space models, as we keep equal parameter count while increasing depth. The memory function takes the form of an impulse

$$\rho(s, \alpha) = \begin{cases} 1 & \text{if } s = \alpha, \\ 0 & \text{otherwise} \end{cases} \tag{15}$$

Here, the parameter $\alpha$ controls the shifting distance. The architecture used in these experiments is identical to the one analyzed in our theoretical framework, i.e. deep linear SSMs. For each depth $l$, we solve for the width $m$ such that the total number of parameters satisfies the relation $l(m-1)+1 = 31$, matching the parameter counts of the one-layer model with width $31$. As shown in Figure 1, we present the mean approximation error together with $\pm 1$ standard deviation across different seeds. We could see that the approximation error decreases with better performance when we try to fix the same expressivity as the number of layers increases. Each point on the additional line in the plot represents the norm of a one-layer linear SSM that is equivalent to the corresponding deep linear SSMs. We find that increasing the number of layers while reducing the width by keeping the total number of effective parameters fixed leads to improved performance. However, this improvement comes with a trade-off: as the number of layers increases, the wall-clock time per epoch becomes slower. Furthermore, we observe that as the depth of the model increases, the corresponding norm required by the equivalent one-layer SSM also increases, which is the similar trend as Theorem 2 predicts. We compute the corresponding norm of one-layer SSM with the same expressivity by Lemma 2. This demonstrates that depth contributes to improved model performance when the parameter count is fixed.

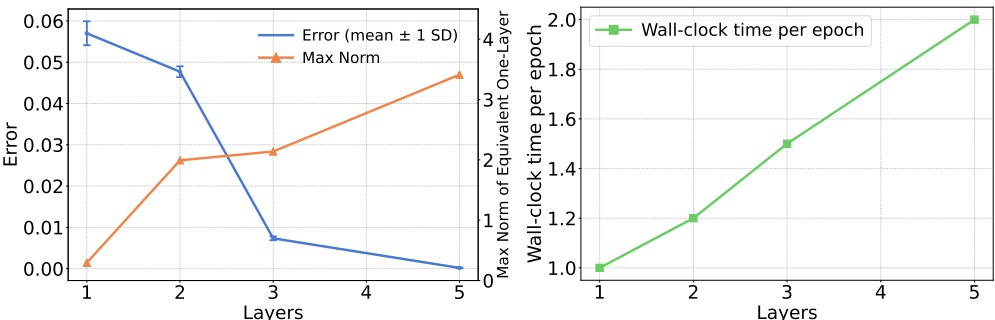

Figure 1: The left panel shows the training error and the maximum norm of the equivalent one-layer SSM computed by $Lemma$ 2 when using linear SSMs of varying depths, all designed to achieve the same expressivity, to learn a linear functional impulse. The right panel reports wall-clock time per epoch of SSMs with different depths, normalized by the runtime of a one-layer linear SSM. We could notice that the data point for depth 4 layers is infeasible because $4 \times (m-1) + 1 = 31$ implies $m = 7.5$ which does not admit a positive integer solution for $m$ and is not valid in our setting where the width must be an integer.

Next, we explain why parameter norm control is important. We train linear SSMs with different depth and width but we keep the same parameter counts to learn the impulse memory functions defined as equation Equation (15) at positions $\alpha = 10$ and $\alpha = 50$ and the results are shown in the Table 1.

The impulse memory function with $\alpha = 10$ is a "small-norm" target whereas that with $\alpha = 50$ is a relatively "large-norm target". By this we mean that to learn the target for $\alpha = 50$ at fixed model sizes, it is known that one requires larger norms in the trainable parameters to reach a similar approximation error from our Proposition 1. All models are trained for approximately the same wall-clock time ($\approx 190$s), so deeper models see fewer optimization steps. For the impulse memory function target at $\alpha = 10$, the shallow 1-layer SSM converges fastest and attains the lowest loss. For the impulse memory function target at $\alpha = 50$ which requires larger parameter norms to reach a similar approximation error, the deepest 3-layer SSM achieves the best loss, even though it runs

Table 1: Training steps and loss under an equal wall-clock budget 190s.

| Architecture | Impulse $\alpha = 10$ | | Impulse $\alpha = 50$ | |
|---|---|---|---|---|
| | Steps (190.72s) | Loss | Steps (190.88s) | Loss |
| 1 layer, width 31 | 3328 | **2.07e − 05** | 3328 | $1.30e − 03$ |
| 2 layers, width 16 | 2195 | $2.28e − 05$ | 2199 | $5.56e − 04$ |
| 3 layers, width 11 | 1603 | $3.36e − 05$ | 1613 | **5.27e − 04** |

slower per step. This is consistent with Theorem 2: depth could help reduce the norm. A deep SSM can realize a large effective kernel norm by factorizing it into a product of smaller per-layer norms, whereas a shallow SSM cannot, making the large-norm target harder to approximate under norm constraints.

## 5.2 EXPERIMENTS ON NONLINEAR S4D MODEL ON SEQUENTIAL MNIST AND CIFAR-10

We aim to validate our theoretical findings on a practical model. To this end, we conduct experiments with the S4D model (Gupta et al., 2022) on the sequential MNIST task (Xiao et al., 2017) and sequential CIFAR-10 task (Tay et al., 2021). We stack linear SSMs in different layers with no internal nonlinearity and adjust depth and width to keep the same parameter count. A single, identical FFN is appended to different stacked linear SSM layers while we keep the FFN fixed so that only the SSM stack differs. Under this nonlinear S4D design, we treat models with the same parameter budget as approximately equivalent. We give an intuitive illustration of the architecture we use in Appendix D.4. The input to this model is a sequence derived from the flattened MNIST image (1D sequence) and the output is the predicted class. When $l = 1$, this reduces to the standard S4D model (Gupta et al., 2022). When $l \geq 1$, we reduce $d_{\text{state}}$ to keep the width fixed, thereby preserving the expressivity of the model.

The relevant theoretical result for our experiment is Theorem 2, which shows that the norm of the weights in deep linear SSMs decreases exponentially in $l^{-1}$ as the number of layers increases. For optimization stability, the parameter norm of each layer would be upper-bounded by some constant, denoted as $B$. Then, according to Theorem 2, the maximum parameter norm of the equivalent one-layer SSM that can be expressed by such an $l$-layer SSM would be upper-bounded by $O(B^{l/2})$, which is exponential in $l$.

For the experiment on the MNIST task (Xiao et al., 2017), each point of the orange line in the left panel of Figure 2 is the maximum norm of weights equivalent to a one-layer nonlinear S4D across different layers while keeping the same parameter count. We can see that the maximum norm of weights equivalent to a one-layer S4D increases exponentially with the number of layers. While Theorem 2 is established in a linear setting, we observe a qualitatively similar trend in Figure 2 for the nonlinear S4D model trained on MNIST: increasing depth leads to a substantial reduction in model norm. The blue line in the left panel of Figure 2 shows that increasing the number of layers, while reducing the hidden dimension to maintain the same expressivity, can lead to improved model performance. Due to the nonlinearity of S4D, this experiment does not provide an exact quantitative match to the theory, but it suggests that the exponential norm-reduction phenomenon may generalize beyond linear models. However, this improvement also comes with the same trade-off: as the number of layers increases, the computational speed becomes slower.

We also conducted experiments using nonlinear S4D models on the sequential CIFAR-10 task (Tay et al., 2021). The results demonstrate that increasing the model depth, while keeping the total parameter count approximately constant, leads to improved test accuracy. Specifically, models with greater depth and narrower width consistently outperform shallower, wider counterparts. The best test accuracy for models with different depth and width is reported in Table 2. Here, we report the best test accuracy across multiple runs in Table 2 to intentionally avoid the influence of optimization error.

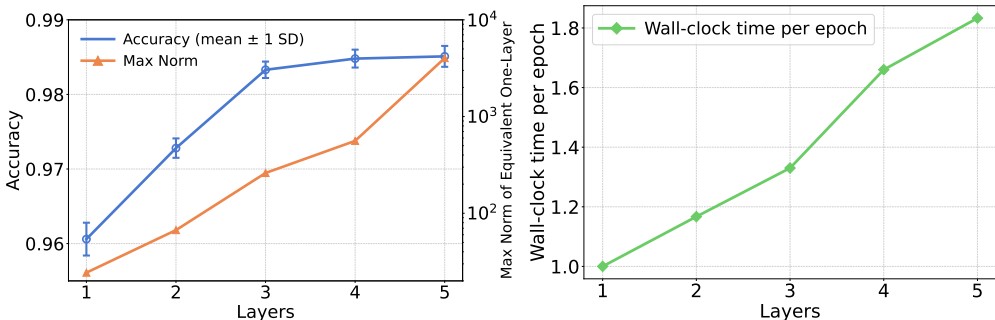

Figure 2: The left panel displays the test accuracy reported as mean $\pm 1$ standard deviation and the maximum norm of the equivalent one-layer computed by Lemma 2 for the S4 model of varying depths on MNIST. All are designed to achieve the same expressivity. The right panel reports the wall-clock time per epoch of SSMs with different depths, normalized by the runtime of a one-layer SSM.

| Layers | Width | Test Accuracy |
|--------|-------|---------------|
| 4 | 64 | 0.8048 |
| 3 | 86 | 0.8036 |
| 2 | 128 | 0.7978 |
| 1 | 256 | 0.7846 |

Table 2: Test accuracies for nonlinear SSMs on sequential CIFAR-10 across layers and width; here layers denote the number of stacked SSM layers.

## 6 CONCLUSION

In this work, we investigate the effect of depth on the expressivity of deep linear state-space models under norm constraints. Our theoretical analyses in Theorem 1 reveal a nontrivial dependence of model expressivity on depth when the total parameter count is held constant. In particular, while increasing depth and width are generally equivalent in the absence of norm constraints, their roles differ significantly under norm-bounded regimes. In Theorem 2, we also show that increasing depth can significantly reduce the required parameter norms under the same parameter count, especially in tasks that demand large-norm representations such as modeling oscillatory or non-smooth memory functions (Jiang et al., 2025; Liu & Li, 2025). This norm reduction highlights the effectiveness of deeper architectures as our experiments shows in Section 5. Moreover, our results suggest a promising direction for future research: factoring shallow SSMs into deeper ones can enhance expressivity and generalization, even in classical implementations. While this may incur a runtime cost due to increased depth, it opens the door to better optimization and stability, especially when norm control is critical. Understanding this trade-off more thoroughly in nonlinear or real-world SSM variants remains an exciting avenue for future exploration. To the best of our knowledge, this is the first rigorous analysis of (norm constrained) capacity of deep linear SSMs, offering theoretical insights into their norm weights and representational abilities as depth increases. Our theoretical results are valuable by revealing the role of depth and width under the parameters norm in deep linear SSMs. It is potentially useful to extend our theoretical framework to designing computational efficient architecture.

**Limitations and Future Work**  This work has several limitations. First, our analysis is confined to the linear setting; although we expect many conclusions to extend to nonlinear models, we do not explicitly treat convolutional representations. Nevertheless, our experiments on a nonlinear architecture provide suggestive evidence that the main insights persist. Second, our discussion of depth is purely approximation-theoretic; once implicit or explicit regularization is taken into account, norm constraints naturally arise and lead to a distinct problem formulation beyond our current scope. Finally, as shown in Section 5.2, wall-clock time per epoch increases with depth, raising practical concerns about computational cost; future work should therefore develop more efficient methods for multi-layer computations to better trade off depth-related gains against training speed.

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

## A  PROOFS OF THE THEOREMS AND LEMMAS

### A.1  PROOFS ON MODEL STRUCTURES AND REPRESENTATIONS

**Proof of Lemma 1**   It is clear that $h_{n_1}(t)$ is linear w.r.t. $x(s)$, for $n_1 = 1, \ldots, l$ and $s = 0, \ldots, t$. Then we denote as $h_{(n_1)}(t) = \sum_{s=0}^{t} h(s, n_1) x(t-s)$. To prove this lemma, we first apply induction on $s$ and $n_1$ to prove the following statement:

$$h(s, n_1) = \sum_{\substack{i_1+i_2+\ldots+i_{n_1}=s \\ i_1,\ldots,i_{n_1} \in \mathbb{N}}} \prod_{j=1}^{n_1} (A_{n_1-j+1}^{i_{n_1-j+1}} B_{n_1-j+1}), \forall n_1 = 1, \ldots, l \tag{16}$$

When $s = 0$, it is clear that $h(0, n_1) = \prod_{j=1}^{n_1} B_{n_1-j+1}$, $\forall n_1$. When $n_1 = 1$, it is clear that $h(s, 1) = A_1^s B_1$, $\forall s$. Then suppose this representation for $h$ is true for $h(s, n_1 - 1)$ and $h(s - 1, n_1)$, then we have

$$h(s, n_1) = A_{n_1} h(s-1, n_1) + B_{n_1} h(s, n_1 - 1) \tag{17}$$

$$= A_{n_1} \sum_{\substack{i_1+i_2+\ldots+i_{n_1}=s-1 \\ i_1,\ldots,i_{n_1} \in \mathbb{N}}} \prod_{j=1}^{n_1} (A_{n_1-j+1}^{i_{n_1-j+1}} B_{n_1-j+1}) \tag{18}$$

$$+ B_{n_1} \sum_{\substack{i_1+i_2+\ldots+i_{n_1}=s \\ i_1,\ldots,i_{n_1-1} \in \mathbb{N}}} \prod_{j=1}^{n_1-1} (A_{n_1-1-j+1}^{i_{n_1-1-j+1}} B_{n_1-1-j+1}) \tag{19}$$

$$= \sum_{\substack{i_1+i_2+\ldots+i_{n_1}=s \\ i_1,\ldots,i_{n_1} \in \mathbb{N} \\ i_{n_1} \geq 1}} \prod_{j=1}^{n_1} (A_{n_1-j+1}^{i_{n_1-j+1}} B_{n_1-j+1}) \tag{20}$$

$$+ \sum_{\substack{i_1+i_2+\ldots+i_{n_1}=s \\ i_1,\ldots,i_{n_1} \in \mathbb{N} \\ i_{n_1} = 0}} \prod_{j=1}^{n_1} (A_{n_1-j+1}^{i_{n_1-j+1}} B_{n_1-j+1}) \tag{21}$$

$$= \sum_{\substack{i_1+i_2+\ldots+i_{n_1}=s \\ i_1,\ldots,i_{n_1} \in \mathbb{N}}} \prod_{j=1}^{n_1} (A_{n_1-j+1}^{i_{n_1-j+1}} B_{n_1-j+1}) \tag{22}$$

Then we have that with the expression of $h(s - 1, n_1)$ and $h(s, n_1 - 1)$, we can have the expression of $h(s, n_1)$ as above. Then with $h(0, n_1)$ and $h(s, 1)$, induction shows the correctness of the above expression. Thus

$$\rho(t) = C^T h(t, l) = C^T \sum_{\substack{i_1+i_2+\ldots+i_l=t \\ i_1,\ldots,i_l \in \mathbb{N}}} \prod_{j=1}^{l} (A_{l-j+1}^{i_{l-j+1}} B_{l-j+1}) \tag{23}$$

$\square$

### A.2  REWRITING OF EXPLICIT FORMS

Before we come to the proof of **Theorem 1**, we prove the following lemmas to aid our proofs.

**Definition 1**   Denote by $F_k(\alpha_1, \ldots, \alpha_n)$ the following function:

$$F_k(\alpha_1, \ldots, \alpha_n) = \sum_{i=1}^{n} \frac{\alpha_i^{k+n-1}}{\prod_{j \neq i} (\alpha_i - \alpha_j)}$$

where $n$ is the number of inputs taken into the function, and $\alpha_i$ are distinct complex numbers.

**Corollary 1** $F_k(\alpha_1, \ldots, \alpha_n) = F_k(\alpha_{\sigma(1)}, \ldots, \alpha_{\sigma(n)})$ where $\sigma$ is a permutation. And $F_k(\alpha_1, \ldots, \alpha_n) = F_k(\alpha_1, \ldots, \alpha_n, 0)$.

The corollary is easy to verify, so we omit the proof here.

**Lemma 3.** For all integer $k$ satisfying $0 \le k \le n-2$, and distinct complex number $\alpha_1, \ldots, \alpha_n$, we have

$$\sum_{i=1}^n \frac{\alpha_i^k}{\prod_{j \neq i}(\alpha_i - \alpha_j)} = 0$$

**Proof of Lemma 3.** For a given integer $1 \le k \le n-2$, consider the following polynomial of $x$ of degree at most $n-1$:

$$f(x) = \sum_{i=1}^n \frac{\alpha_i^k \prod_{j \neq i}(x - \alpha_j)}{\prod_{j \neq i}(\alpha_i - \alpha_j)}$$

It is obvious by the properties of Lagrange Interpolation Polynomial that $f(x) = x^k$, thus the coefficient of $x^{n-1}$ is 0, this gives exactly

$$\sum_{i=1}^n \frac{\alpha_i^k}{\prod_{j \neq i}(\alpha_i - \alpha_j)} = 0$$

$\square$

**Lemma 4.** Denote by $I(n, k)$ the following set

$$I(n, k) = \{(i_1, \ldots, i_n) : \sum_{j=1}^n i_j = k; \quad i_j \ge 0, i_j \in \mathbb{Z}, \forall j\}$$

Then for distinct $\alpha_1, \ldots, \alpha_n$, we have

$$\sum_{(i_1, \ldots, i_n) \in I(n,k)} \left(\prod_{j=1}^n \alpha_j^{i_j}\right) = F_k(\alpha_1, \ldots, \alpha_n)$$

**Proof of Lemma 4.** We do induction over $n$. It is easy to calculate that when $n = 1, \forall k \ge 0$,

$$\sum_{(i_1, \ldots, i_n) \in I(n,k)} \left(\prod_{j=1}^n \alpha_j^{i_j}\right) = \alpha_1^k = F_k(\alpha_1)$$

Now suppose that for $n-1$ and $\forall k \geq 0$, the equalities holds. Then we consider the case of $n$.

$$\sum_{(i_1,\ldots,i_n) \in I(n,k)} (\prod_{j=1}^{n} \alpha_j^{i_j}) = \sum_{k_1=0}^{k} \alpha_n^{k-k_1} (\sum_{(i_1,\ldots,i_{n-1}) \in I(n-1,k_1)} (\prod_{j=1}^{n-1} \alpha_j^{i_j})) \tag{24}$$

$$= \sum_{k_1=0}^{k} \alpha_n^{k-k_1} F_{k_1}(\alpha_1,\ldots,\alpha_{n-1}) \tag{25}$$

$$= \sum_{k_1=0}^{k} \sum_{i=1}^{n-1} \frac{\alpha_i^{k_1+n-2} \alpha_n^{k-k_1}}{\prod_{j \neq i; j \neq n}(\alpha_i - \alpha_j)} \tag{26}$$

$$= \sum_{i=1}^{n-1} \frac{\alpha_i^{k+n-1} - \alpha_n^{k+1} \alpha_i^{n-2}}{(\prod_{j \neq i; j \neq n}(\alpha_i - \alpha_j))(\alpha_i - \alpha_n)} \tag{27}$$

$$= \sum_{i=1}^{n-1} \frac{\alpha_i^{k+n-1} - \alpha_n^{k+1} \alpha_i^{n-2}}{\prod_{j \neq i}(\alpha_i - \alpha_j)} \tag{28}$$

$$= \sum_{i=1}^{n-1} \frac{\alpha_i^{k+n-1}}{\prod_{j \neq i}(\alpha_i - \alpha_j)} + \alpha_n^{k+1}(0 - \sum_{i=1}^{n-1} \frac{\alpha_i^{n-2}}{\prod_{j \neq i}(\alpha_i - \alpha_j)}) \tag{29}$$

$$= \sum_{i=1}^{n-1} \frac{\alpha_i^{k+n-1}}{\prod_{j \neq i}(\alpha_i - \alpha_j)} + \alpha_n^{k+1}(\frac{\alpha_n^{n-2}}{\prod_{j \neq n}(\alpha_n - \alpha_j)}) \tag{30}$$

$$= F_k(\alpha_1,\ldots,\alpha_n) \tag{31}$$

Where the second last step uses the result of **Lemma 3**. $\qquad\square$

As the set $\{(\alpha_1,\ldots,\alpha_n) : \alpha_1,\ldots,\alpha_n \text{ are distinct}\}$ is open and dense in $\mathbb{C}^n$, and taking finite sum in the form of $\sum_{(i_1,\ldots,i_n) \in I(n,k)}(\prod_{j=1}^{n} \alpha_j^{i_j}) = F_k(\alpha_1,\ldots,\alpha_n)$ is continuous w.r.t. $(\alpha_1,\ldots,\alpha_n)$ under norms on $\mathbb{C}^n$, thus we can extend the definition of $F_k(\alpha_1,\ldots,\alpha_n)$ to the entire $\mathbb{C}^n$ by taking limits.

**Corollary 2.** For a $l$ layer SSM $\rho \in \mathcal{H}_{\infty,l}^m$, suppose it is defined by matrix $A_i, i = 1,\ldots,l$; $B_i, i = 1,\ldots,l$ and $C$ as given in the formulation 1. Then we have

$$\rho(t) = \sum_{1 \leq j_1,\ldots,j_l \leq m} C_{j_l}[\prod_{p=1}^{l-1}(B_{l-p+1})_{(j_{l-p+1},j_{l-p})}](B_1)_{j_1}[\sum_{\substack{i_1+i_2+\ldots+i_l=t \\ i_1,\ldots,i_l \in \mathbb{N}}} \prod_{j=1}^{l}((A_{l-j+1})_{(j_{l-p+1},j_{l-p+1})}^{i_{l-j+1}})] \tag{32}$$

If we denote $\alpha(x,y)$ as $\alpha(x,y) = (A_x)_{(y,y)}$ for diagonal matrices $A_x$, then we have

$$\rho(t) = \sum_{1 \leq j_1,\ldots,j_l \leq m} C_{j_l}[\prod_{p=1}^{l-1}(B_{l-p+1})_{(j_{l-p+1},j_{l-p})}](B_1)_{j_1} F_t(\alpha(1,j_1),\ldots,\alpha(l,j_l)) \tag{33}$$

This corollary is straight from calculations of **Lemma 1**

**Lemma 5.** Given fixed positive integers $M \geq N, T > 0$. Denote by $B(\alpha,\epsilon) = \{c \in \mathbb{C} : |c - \alpha| < \epsilon\}$ for real number $\epsilon > 0$. For fixed complex numbers $b_1,\ldots,b_N$ satisfying $\sum_{i=1}^{N} b_i \neq 0$, and pairwise distinct non-zero complex numbers $\alpha_1,\ldots,\alpha_n$. Then for $T > N + M + 1$ and sufficiently small $\epsilon$, we have

$$\lim_{\epsilon \to 0}(\inf_{\substack{\beta_1,\ldots,\beta_M \in \cup_{i=1}^{N} B(\alpha_i,\epsilon) \\ \sum_{i=1}^{M} c_i = 0}} \sum_{t=1}^{T} |(\sum_{i=1}^{N} b_i \alpha_i^t) - (\sum_{i=1}^{M} c_i \beta_i^t)|) > f(\{\alpha_i\},\{b_i\}) > 0 \tag{34}$$

Where $f(\{\alpha_i\},\{b_i\})$ is a real number independent of $\epsilon$.

**Proof of Lemma 5.** It is clear that for sufficiently small $\epsilon$, $\beta_i$ have to be non-zero, and $B(\alpha_i, \epsilon)$ are disjoint. Then we rewrite $(\sum_{i=1}^M c_i \beta_i^t)$ as

$$\sum_{i=1}^M c_i \beta_i^t = \sum_{j=1}^N \sum_{\beta_{ji} \in B(\alpha_j, \epsilon)} c_{ji} \beta_{ji}^t \tag{35}$$

Then by considering the first-order asymptotic of $\sum_{\beta_{ji} \in B(\alpha_j, \epsilon)} c_{ji} \beta_{ji}^t$ w.r.t. $\epsilon$, we have that

$$\sum_{\beta_{ji} \in B(\alpha_j, \epsilon)} c_{ji} \beta_{ji}^t = B_j(t) \alpha_j^t + o(1) \tag{36}$$

Where $B_j(t)$ are polynomials with $\sum_{j=1}^N deg(B_j) \le M - N$ (The possible existence of polynomial comes from cases where $\sum_{\beta_{ji} \in B(\alpha_j, \epsilon)} c_{ji} = 0$, leading to cancellation of the originally highest order term. In the first order asymptotic we consider the highest order not canceled out in the summation).

If $\sum_{j=1}^N deg(B_j) \ne 0$, then it is clear that such $f(\{\alpha_i\}, \{b_i\})$ exists, as at least one $B_j(t)$ grows at least linearly w.r.t. $t$ but the corresponding $b_j$ remains constant.

If $\sum_{j=1}^N deg(B_j) = 0$, then there's no highest-order cancellation in the way above (canceling the entire term doesn't count in this case). Then by writing $B_j(t) = B_j$, then minimum becomes

$$\lim_{\epsilon \to 0} \left( \inf_{\sum_{i=1}^M B_i = 0} \sum_{t=1}^T |(\sum_{i=1}^N b_i \alpha_i^t) - (\sum_{i=1}^N B_i \alpha_i^t)| + O(\epsilon) \right) \tag{37}$$

And it is clear from $\sum_{i=1}^N b_i \ne 0$ that such limit is strictly greater than zero. Thus the $f(\{\alpha_i\}, \{b_i\})$ exists. $\qquad\square$

### A.3 Proofs of Main Theorems

**Lemma 6.** $\mathcal{H}_{\infty,1}^{l(m-1)+2} \not\subseteq \mathcal{H}_{\infty,l}^m$.

**Proof of Lemma 6.** We consider $\rho \in \mathcal{H}_{\infty,1}^{l(m-1)+2}$ defined by matrices $A_0, B_0, C_0$, where $A_0 = Diag\{\sigma_1, \ldots, \sigma_K\}$ with $K = l(m-1)+2$ and distinct nonzero $\sigma_i$'s. Then we have $\rho(t) = \sum_{i=1}^K (B_0)_i (C_0)_i \sigma_i^t$. Assume the contrary, if $\rho(t) \in \mathcal{H}_{\infty,l}^m$, then suppose it is defined by $A_1, \ldots, A_l$; $B_1, \ldots, B_l$; $C$ as in the formulation, and denote $\alpha(x, y) = (A_x)_{(y,y)}$ to be the diagonal elements of $A_x$. According to the formulation in **Corollary 2.**, it is clear that $\{\alpha(i,j) : 1 \le i \le l, 1 \le j \le m\} = \{\sigma_i : 1 \le i \le K\} \cup \{0\}$.

If there are any two non-zero $\alpha(x_1, y_1) = \alpha(x_2, y_2)$, then we perturb them up to the magnitude of $\epsilon$ such that all the non-zero $\alpha(x, y)$ are distinct, and satisfying that $\alpha(x, y) \in (\cup_{i=1}^K B(\sigma_i, \epsilon)) \cup \{0\}$. As $\epsilon \to 0$, the perturbation of $\rho(t)$ also goes to 0 for any fixed $t$.

Now we consider $F_t(\alpha(1, j_1), \ldots, \alpha(l, j_l))$, and eliminate all the zeros according to the rule in **Corollary 1.**. Then we denote that $F_t(\alpha(1, j_1), \ldots, \alpha(l, j_l)) = F_t(\beta_1, \ldots, \beta_{l_1})$, where $l_1 \ge 2$ due to that there's at most $l - 2$ zeros, and $\beta_i$ are all the non-zero elements in $\{\alpha(i, j_i)\}$, and $\beta_i$ are distinct due to that non-zero $\{\alpha(i, j_i)\}$ are perturbed to be distinct.

Recalling the definition, we have

$$F_t(\beta_1, \ldots, \beta_{l_1}) = \sum_{i=1}^{l_1} \frac{\beta_i^{t+l_1-1}}{\prod_{j \ne i}(\beta_i - \beta_j)} \tag{38}$$

$$= \sum_{i=1}^{l_1} \left( \frac{\beta_i^{l_1-2}}{\prod_{j \ne i}(\beta_i - \beta_j)} \right) \beta_i^{t+1} \tag{39}$$

where the sum of coefficients is

$$\sum_{i=1}^{l_1} \left( \frac{\beta_i^{l_1-2}}{\prod_{j \ne i}(\beta_i - \beta_j)} \right) = 0 \tag{40}$$

As according to **Lemma 3**. And this applies for all such $F_t(\alpha(1, j_1), \ldots, \alpha(l, j_l))$. Then by taking $N = K$, $M = lm$, $b_i = \frac{(B_0)_i(C_0)_i}{\sigma_i}$, $\alpha_i = \sigma_i$, and $\beta_j = \alpha(x, y)$ (with the perturbation applied), then we have exactly the setting of **Lemma 5**.

Then according to **Lemma 5**, we have that for the perturbed $\hat{\rho}(t)$, $\sum_{t=1}^{2ml+1} |\rho(t) - \hat{\rho}(t)| > f_0 > 0$, where $f_0$ is completely determined by the original system defining $\rho$ and independent of the magnitude of perturbation $\epsilon$.

However, we know that for any bounded range of $t$, the summation $\sum_{\substack{i_1+i_2+\ldots+i_l=t \\ i_1,\ldots,i_l \in \mathbb{N}}} \prod_{j=1}^{l}((A_{l-j+1})_{(j_{l-p+1}, j_{l-p+1})}^{i_{l-j+1}})$ in **Corollary 2** has perturbation bounded by a function of $\epsilon$ that goes to zero for $\epsilon \to 0$. This is in contradiction with the positive bound $f_0$. Thus concludes the proof. $\square$

**Lemma 7.** $\mathcal{H}_{\infty,l}^{m} \subseteq \mathcal{H}_{\infty,1}^{lm}$.

**Proof of Lemma 7.** For $\rho \in \mathcal{H}_{\infty,l}^{m}$ defined by matrices $A_i, i = 1, \ldots, l$; $B_i, i = 1, \ldots, l$; $C$ according to the formulation, we construct a 1 layer SSM of width $ml$ defined by vector $C_0, B_0 \in \mathbb{C}^{ml \times 1}$ and $A_0 \in \mathbb{C}^{ml \times ml}$, such that the hidden state $f^{(t)}$ at time $t$ is exactly $(h_1^T(t), \ldots, h_l^T(t))^T$.

We denote $A_0 = (A_0)_{(ij)}, 1 \le i, j \le l$, where each $(A_0)_{(ij)}$ is a $m \times m$ block. And $B_0 = (\beta_1, \ldots, \beta_l)$, $C_0 = (c_1, \ldots, c_l)$ are separated into $l$ of $m \times 1$ vectors stacked together.

$$(A_0)_{(ij)} = \begin{cases} (\prod_{k=1}^{i-j} B_{i+1-k})A_j, i > j \\ A_i, i = j \\ 0, i < j \end{cases} \quad (41)$$

And $b_i = \prod_{j=1}^{i} B_{i+1-j}$, $C_0 = (0, \ldots, 0, C)$. By direct calculation, we could verify that the $\hat{\rho}$ corresponding to the SSM defined by $A_0, B_0, C_0$ is exactly $\rho$, which means $\rho \in \mathcal{H}_{\infty,1}^{lm}$. $\square$

**Corollary 3.** For distinct complex numbers $\gamma_1, \ldots, \gamma_n$, we have that for non-negative integer $t$

$$F_t(\gamma_1, \ldots, \gamma_n) - \frac{\gamma_{n-1}}{\gamma_{n-1} - \gamma_n} F_t(\gamma_1, \ldots, \gamma_{n-1}) = \frac{\gamma_n}{\gamma_n - \gamma_{n-1}} F_t(\gamma_1, \ldots, \gamma_{n-2}, \gamma_n) \quad (42)$$

This result in **Corollary 3** could be derived directly by expanding $F_t$ according to **Definition 1** and simple calculations.

**Lemma 8.** Given $n$ distinct non-zero complex number $\beta_1, \ldots, \beta_n$, and $n$ complex number $Z_1, \ldots, Z_n$. If we define $H_k, k = 1, \ldots, n$ to be

$$H_k = \sum_{j=k}^{n} Z_j \frac{\beta_k \prod_{p=1}^{k-1}(\beta_j - \beta_p)}{\beta_j^k} \quad (43)$$

Then we have

$$\sum_{k=1}^{n} H_k F_t(\beta_1, \ldots, \beta_k) = \sum_{k=1}^{n} Z_k \beta_k^t \quad (44)$$

Furthermore, if $|\beta_i|$ is in an non-decreasing order, then we have $|H_k| \le 2^n \max_{1 \le i \le n} |Z_i|, \forall k$.

**Proof of Lemma 8.** We use induction to prove the following stronger result for all $u = 1, \ldots, n$:

$$\sum_{k=u}^{n} H_k F_t(\beta_1, \ldots, \beta_k) = \sum_{k=u}^{n} Z_k \frac{\prod_{p=1}^{u-1}(\beta_k - \beta_p)}{\beta_k^{u-1}} F_t(\beta_1, \ldots, \beta_{u-1}, \beta_k) \quad (45)$$

In particular, when $u = 1$, it becomes exactly the result we want.

When $u = n$, it is clear that the above is true. We then calculate the following for $u \geq 2$:

$$\sum_{k=u-1}^{n} Z_k \frac{\prod_{p=1}^{u-2}(\beta_k - \beta_p)}{\beta_k^{u-2}} F_t(\beta_1, \ldots, \beta_{u-2}, \beta_k) - \sum_{k=u}^{n} Z_k \frac{\prod_{p=1}^{u-1}(\beta_k - \beta_p)}{\beta_k^{u-1}} F_t(\beta_1, \ldots, \beta_{u-1}, \beta_k)$$
(46)

$$= Z_{u-1} \frac{\prod_{p=1}^{u-2}(\beta_{u-1} - \beta_p)}{\beta_{u-1}^{u-2}} F_t(\beta_1, \ldots, \beta_{u-2}, \beta_{u-1})$$
(47)

$$+ \sum_{k=u}^{n} Z_k \left( \frac{\prod_{p=1}^{u-2}(\beta_k - \beta_p)}{\beta_k^{u-2}} F_t(\beta_1, \ldots, \beta_{u-2}, \beta_k) - \frac{\prod_{p=1}^{u-1}(\beta_k - \beta_p)}{\beta_k^{u-1}} F_t(\beta_1, \ldots, \beta_{u-1}, \beta_k) \right)$$
(48)

$$= Z_{u-1} \frac{\prod_{p=1}^{u-2}(\beta_{u-1} - \beta_p)}{\beta_{u-1}^{u-2}} F_t(\beta_1, \ldots, \beta_{u-2}, \beta_{u-1})$$
(49)

$$+ \sum_{k=u}^{n} Z_k \frac{\prod_{p=1}^{u-1}(\beta_k - \beta_p)}{\beta_k^{u-1}} \left( \frac{\beta_k}{(\beta_k - \beta_{u-1})} F_t(\beta_1, \ldots, \beta_{u-2}, \beta_k) - F_t(\beta_1, \ldots, \beta_{u-1}, \beta_k) \right)$$
(50)

$$= Z_{u-1} \frac{\prod_{p=1}^{u-2}(\beta_{u-1} - \beta_p)}{\beta_{u-1}^{u-2}} F_t(\beta_1, \ldots, \beta_{u-2}, \beta_{u-1})$$
(51)

$$+ \sum_{k=u}^{n} Z_k \frac{\prod_{p=1}^{u-1}(\beta_k - \beta_p)}{\beta_k^{u-1}} \left( \frac{\beta_{u-1}}{\beta_k - \beta_{u-1}} F_t(\beta_1, \ldots, \beta_{u-2}, \beta_{u-1}) \right)$$
(52)

$$= \sum_{k=u-1}^{n} Z_k \frac{\beta_{u-1} \prod_{p=1}^{u-2}(\beta_k - \beta_p)}{\beta_k^{u-1}} \left( F_t(\beta_1, \ldots, \beta_{u-2}, \beta_{u-1}) \right)$$
(53)

$$= H_{u-1} F_t(\beta_1, \ldots, \beta_{u-2}, \beta_{u-1})$$
(54)

Then it is clear that the stronger result holds for all $u = 1, \ldots, n$. Thus taking $u = 1$ we have the desired result.

If the non-decreasing order of $|\beta_i|$ holds, then we have that

$$|H_k| = \left| \sum_{j=k}^{n} Z_j \frac{\beta_k \prod_{p=1}^{k-1}(\beta_j - \beta_p)}{\beta_j^k} \right|$$
(55)

$$\leq \sum_{j=k}^{n} |Z_j| * \left| \frac{\beta_k}{\beta_j} \right| * \prod_{p=1}^{k-1} \left| \frac{(\beta_j - \beta_p)}{\beta_j} \right|$$
(56)

$$\leq \sum_{j=k}^{n} |Z_j| * 2^{k-1}$$
(57)

$$\leq 2^n \max_{1 \leq i \leq n} |Z_i|$$
(58)

$\square$

**Lemma 9.** $\mathcal{H}_{\infty,1}^{l(m-1)+1} \subseteq \mathcal{H}_{\infty,l}^{m}$.

**Proof of Lemma 9.** For simplicity we denote $K = l(m-1) + 1$.

Then for $\rho \in \mathcal{H}_{\infty,1}^{l(m-1)+1}$, suppose that it is defined by vectors $B_0, C_0$, and matrix $A_0 = Diag\{\sigma_1, \ldots, \sigma_K\}$. Then from **Corollary 2** we have $\rho(t) = \sum_{i=1}^{K}(C_0)_i(B_0)_i\sigma_i^t$.

As the case of $\sigma_i = \sigma_j$ or $\sigma_i = 0$ will lead to the degenerate case, which is weaker than the non-degenerate case, thus we do not consider them here. Now we assume that $\sigma_i$ are distinct and non-zero. Due to the symmetry of the above form, we can assume without loss of generality that $|\sigma_i|$ is in an non-decreasing order.

we define the following functions:

$$\alpha(i,j) = \begin{cases} \sigma_j, & \text{for } i=1; j=1,\ldots,m \\ \sigma_{(i-1)(m-1)+j+1}, & \text{for } i=2,\ldots,l; j=1,\ldots,m \\ 0, & \text{for } i=2,\ldots,l; j=m \end{cases} \tag{59}$$

$$Z(i,j) = \begin{cases} (B_0)_j(C_0)_j, & \text{for } i=1; j=1,\ldots,m \\ (B_0)_{(i-1)(m-1)+j+1}(C_0)_{(i-1)(m-1)+j+1}, & \text{for } i=2,\ldots,l; j=1,\ldots,m \\ 0, & \text{for } i=2,\ldots,l; j=m \end{cases} \tag{60}$$

$$H(i,j) = \begin{cases} \sum_{p=i}^{l} Z(p,j)\frac{\alpha(i,j)\prod_{q=1}^{i-1}(\alpha(p.j)-\alpha(q,j))}{\alpha(p,j)^i}, & \text{for } i=1,\ldots,l; j=1,\ldots,m-1 \\ Z(1,m), & \text{for } i=1; j=m \\ 0, & \text{for } i=2,\ldots,l; j=m \end{cases} \tag{61}$$

Also, denote by $Z_0 = 2 * (\max_{1\le i\le K} |(B_0)_i(C_0)_i|)^{\frac{1}{l+1}}$.

Then we construct diagonal matrices $A_1,\ldots,A_l$; matrices $B_2,\ldots,B_l$; vectors $B_1, C$ such that the $l$ layer SSM of width $m$ defined by these parameters has exactly the same $\rho$.

- $B_0 = C = Z_0 1_m$, where $1_m$ is the vector consisting of all 1.
- $(A_i)_{(j,j)} = \alpha(i,j)$, for $i=1,\ldots,l; j=1,\ldots,m$.
- $(B_i)_{(j,j)} = Z_0$, for $i=2,\ldots,l-1; j=1,\ldots,m-1$.
- $(B_i)_{(m,m)} = Z_0$, for $i=3,\ldots,l$.
- $(B_2)_{(m,m)} = \frac{H(1,m)}{Z_0^l}$.
- $(B_l)_{(j,j)} = \frac{H(l,j)}{Z_0^l}$, for $j=1,\ldots,m-1$.
- $(B_i)_{(m,j)} = \frac{H(i-1,j)}{Z_0^l}$, for $i=2,\ldots,l; j=1,\ldots,m-1$
- All the unmentioned elements are set to zero.

Under such construction, we suppose it defines $\hat{\rho}$, then following **Corollary 2** and **Lemma 8**, we have

$$\hat{\rho}(t) = \sum_{1\le j_1,\ldots,j_l\le m} C_{j_l}[\prod_{p=1}^{l-1}(B_{l-p+1})_{(j_{l-p+1},j_{l-p})}](B_1)_{j_1} F_t(\alpha(1,j_1),\ldots,\alpha(l,j_l)) \tag{62}$$

$$= Z(1,m)\alpha(1,m)^t + \sum_{j=1}^{m-1}\sum_{i=1}^{l} H(i,j)F_t(\alpha(1,j),\ldots,\alpha(i,j)) \tag{63}$$

$$= Z(1,m)\alpha(1,m)^t + \sum_{j=1}^{m-1}\sum_{i=1}^{l} Z(i,j)\alpha(i,j)^t \tag{64}$$

$$= \sum_{i=1}^{K}(C_0)_i(B_0)_i\sigma_i^t \tag{65}$$

$$= \rho(t) \tag{66}$$

Thus $\rho = \hat{\rho} \in \mathcal{H}_{\infty,l}^m$. $\qquad\square$

**Remark 1.** : As for the degenerated cases, as there are less parameters in $\rho_t$, its construction could be achieved by simple modifications on the above construction, or simply achieved by taking limits as the set of non-degenerate $(\sigma_1,\ldots,\sigma_K)$ is open and dense in $\mathbb{C}^K$.

**Remark 2.** : It can be seen from **Lemma 8** that all the entries of $B_i, C$ are bounded by $Z_0 = 2 * (\max_{1\le i\le K} |(B_0)_i(C_0)_i|)^{\frac{1}{l+1}}$. As for the degenerated cases, we can do the construction by taking limits, thus for these cases the same bound holds.

**Proof of Theorem 1**    This is direct from **Lemma 6**, **Lemma 7**, **Lemma 9**. $\qquad\square$

**Proof of Theorem 2**  For $\rho \in \mathcal{H}^{l(m-1)+1}_{c_1,1}$, we have that

$$Z_0 = 2 * (\max_{1 \leq i \leq K} |(B_0)_i (C_0)_i|)^{\frac{1}{l+1}} \leq 2c_1^{\frac{2}{l+1}} \tag{67}$$

Then it is direct from **Lemma 9, Remark 2**. $\qquad\square$

**Proof of Theorem 3**  From **Theorem 2**, it suffices that

$$2c_1^{\frac{2}{l+1}} \leq c_2 \tag{68}$$

Which means

$$l \geq \lceil \frac{2\ln(c_1)}{\ln\left(\frac{c_2}{2}\right)} - 1 \rceil \tag{69}$$

For this $l$, we already have that $m = \lceil \frac{K}{l} \rceil + 1$ satisfies that $K \leq l(m-1)$, and **Lemma 9, Remark 2** shows that the bound also holds for degenerate cases, which is valid to be used here. $\qquad\square$

**Proof of Corollary 1**  To prove this corollary, we only need to show that $\mathcal{G}^{l(m-1)+1}_{c_1,1} \subseteq \mathcal{H}^{l(m-1)+1}_{\sqrt{l(m-1)+1}c_1,1}$. Assume $\rho \in \mathcal{G}^{l(m-1)+1}_{c_1,1}$ is defined by $A_1, B_1, C$, and based on the property of normal matrices, we can decompose as $A_1 = U\Sigma U^*$ for diagonal $\Sigma$ and unitary $U$. Then we have

$$\rho(t) = C^T A_1^t B_1 \tag{70}$$

$$= C^T U \Sigma^t U^* B_1 \tag{71}$$

$$= (U^T C)^T \Sigma^t (U^* B_1) \tag{72}$$

Therefore $\rho$ is also defined by $\Sigma, (U^T C), (U^* B_1)$. And by the norm-preserving property of unitary matrix, we have $\|(U^T C)\|_2 = \|C\|_2, \|U^* B_1\|_2 \leq \|B_1\|_2$. Then $\|(U^T C)\|_\infty \leq \sqrt{l(m-1)+1}c_1$, $\|(U^* B_1)\|_\infty \leq \sqrt{l(m-1)+1}c_1$, leading to $\rho \in \mathcal{G}^{l(m-1)+1}_{\sqrt{l(m-1)+1}c_1,1}$. Then by **Theorem 2**, the conclusion holds. $\qquad\square$

**Proof of Lemma 2**  This lemma can be directly obtained from **Corollary 2** by indexing $F_t(\alpha(1, j_1), \ldots, \alpha(l, j_l))$ as according to **Definition 1**. $\qquad\square$

**Remark 3.**  We could define an even larger space as follows:

$$\mathcal{L}^m_{c,l} = \{\rho(t) : y(t) = (\rho * x)(t), A_1, ..., A_l \in \mathbb{C}^{m \times m} \text{ diagonalizable}, B_2, ..., B_l, \in \mathbb{C}^{m \times m}, \tag{73}$$

$$C, B_1 \in \mathbb{C}^{m \times 1}, \max_{i=1,...,l} r(A_i) < 1, \|C\|_\infty \leq c, \|B_1\|_\infty \leq c, \max_{2 \leq k \leq l} \max_{1 \leq i,j \leq m} |(B_k)_{ij}| \leq c\} \tag{74}$$

The difference between $\mathcal{H}^m_{c,l}$ and $\mathcal{L}^m_{c,l}$ is not as substantial as one might initially expect. In fact, if we remove the norm constraint-that is, as $c$ approaches infinity-the two hypothesis spaces become identical.

**Lemma 3** (Representing ability equivalence in two hypothesis space).  *Given $m, l \geq 1$, the we have*

$$\mathcal{L}^m_{\infty,l} = \mathcal{H}^m_{\infty,l} \tag{75}$$

**Proof of Lemma 3**  As it is clear that

$$\mathcal{H}^m_{\infty,l} \subseteq \mathcal{L}^m_{\infty,l}$$

. We only need to prove that $\forall \rho \in \mathcal{L}^m_{\infty,l}$, we have $\rho \in \mathcal{H}^m_{\infty,l}$. Suppose this $l$ layer SSM is defined by matrix $A_i, i = 1, \ldots, l$; $B_i, i = 1, \ldots, l$ and $C$ as given in the formulation. Then suppose $A_i = P_i^{-1} D_i P_i$ is the Jordan decomposition of $A_i$, where $P_i$ is invertible and $D_i$ is diagonal. Then

let $\hat{C} = P_l^T C$, $\hat{B}_j = P_j^{-1} B_j P_{j-1}, j = 2, \ldots, l$, $\hat{B}_1 = P_1^{-1} B_1$, $\hat{A}_i = D_i, i = 1, \ldots, l$. It is clear that for all $t$,

$$\rho(t) = C^T \sum_{\substack{i_1+i_2+\ldots+i_l=t \\ i_1,\ldots,i_l \in \mathbb{N}}} \prod_{j=1}^{l} (A_{l-j+1}^{i_{l-j+1}} B_{l-j+1}) \tag{76}$$

$$= \hat{C}^T \sum_{\substack{i_1+i_2+\ldots+i_l=t \\ i_1,\ldots,i_l \in \mathbb{N}}} \prod_{j=1}^{l} (\hat{D}_{l-j+1}^{i_{l-j+1}} \hat{B}_{l-j+1}) \tag{77}$$

As $D_i$ are diagonal matrices, we have that $\rho \in \mathcal{H}_{\infty,l}^m$. $\qquad\square$

### A.4  PROOF OF LEMMA 2

This lemma can be directly obtained from **Corollary 2** by indexing $F_t(\alpha(1, j_1), \ldots, \alpha(l, j_l))$ as according to **Definition 1**.

## B  PROOF OF PROPOSITION

*Proof of Proposition 1.* Fix $m \in \mathbb{N}$ and a diagonal matrix

$$A_1 = \text{diag}(\alpha_1, \ldots, \alpha_m)$$

with spectral radius $r(A_1) < 1$, and let $B = (1, \ldots, 1)^\top \in \mathbb{C}^m$. For $C = (c_1, \ldots, c_m)^\top \in \mathbb{C}^m$, the corresponding one-layer linear SSM (Equation (1) with $l = 1$ and $\sigma = \text{id}$) has convolution kernel (by Lemma 1 applied with $l = 1$)

$$\rho_C(t) = C^\top A_1^t B = \sum_{j=1}^{m} c_j \alpha_j^t, \qquad t \geq 0.$$

Let $k \geq 0$ and $0 < \epsilon < 1$ be fixed, and suppose that $C \in \mathcal{C}_{m,k}(\epsilon)$, i.e.

$$\max_{t \geq 0} |\rho_C(t) - \rho_k(t)| \leq \epsilon, \quad \rho_k(t) = \mathbf{1}_{\{t=k\}}.$$

In particular, evaluating at $t = k$ gives

$$|\rho_C(k) - 1| = |\rho_C(k) - \rho_k(k)| \leq \epsilon \implies |\rho_C(k)| \geq 1 - \epsilon. \tag{78}$$

On the other hand, let

$$r := r(A_1) = \max_{1 \leq j \leq m} |\alpha_j| < 1.$$

For any $t \geq 0$ we have, by the triangle inequality,

$$|\rho_C(t)| = \left| \sum_{j=1}^{m} c_j \alpha_j^t \right| \leq \sum_{j=1}^{m} |c_j| |\alpha_j|^t \leq r^t \sum_{j=1}^{m} |c_j| \leq r^t \, m \, \|C\|_\infty,$$

where $\|C\|_\infty = \max_{1 \leq j \leq m} |c_j|$. Specializing to $t = k$ yields

$$|\rho_C(k)| \leq m \, r^k \, \|C\|_\infty. \tag{79}$$

Combining equation 78 and equation 79 gives

$$1 - \epsilon \leq |\rho_C(k)| \leq m \, r^k \, \|C\|_\infty,$$

hence

$$\|C\|_\infty \geq \frac{1-\epsilon}{m \, r^k}. \tag{80}$$

By the assumption of the proposition, the impulse position $k$ grows at least linearly with the width $m$, i.e. there exists $\beta > 0$ such that

$$k \geq \beta m.$$

Since $0 < r < 1$, the map $k \mapsto r^k$ is decreasing, so from equation 80 we obtain

$$\|C\|_\infty \;\geq\; \frac{1-\epsilon}{m\, r^{\beta m}} = \frac{1-\epsilon}{m} \exp\!\big(\beta m \log(1/r)\big).$$

Let $\lambda := \beta \log(1/r) > 0$. Then

$$\log \|C\|_\infty \;\geq\; \lambda m - \log m + \log(1-\epsilon).$$

The term $-\log m$ grows only logarithmically, while $\lambda m$ grows linearly in $m$, so there exists $m_0 \in \mathbb{N}$ such that for all $m \geq m_0$,

$$\lambda m - \log m + \log(1-\epsilon) \;\geq\; \frac{\lambda}{2}\, m.$$

Equivalently, for all $m \geq m_0$,

$$\|C\|_\infty \;\geq\; \exp\!\Big(\frac{\lambda}{2}\, m\Big).$$

Setting

$$\gamma := \frac{\lambda}{2} = \frac{\beta}{2} \log \frac{1}{r(A_1)} > 0,$$

we conclude that, whenever $k \geq \beta m$ and $C \in \mathcal{C}_{m,k}(\epsilon)$, the coefficients must satisfy

$$\|C\|_\infty \;\geq\; \exp(\gamma m)$$

for all sufficiently large $m$. This proves the claimed exponential growth of the parameter norm with respect to the width $m$ when the impulse position scales linearly with the width. $\qquad\square$

## C  Numerical verification of Theorem 2

We conduct numerical experiments to verify our main theorems. Specifically, given a one-layer linear SSM with width $l(m-1)+1$, we apply the construction from Theorem 2 to reconstruct an equivalent model with $l$ layers and width $m$ where a wide and shallow network is learned using a deep linear network as defined in Equation (1). We use a one-layer deep linear state-space model (SSM) with width 2521 as our target model and use deep linear SSMs with varying depths ranging from 2 to 63 while controlling the same parameter counts across different depths to reconstruct the target model. For each depth $l$, we adjust the width $d$ such that the total number of parameters matches that of the target model. This results in the relation $2521 = l(m-1)+1$. For example, a deep linear SSM model with depth 63 would have width 41 to match the parameter count of the target model. The experiments are performed for both real-valued and complex-valued parameters. In the plot Figure 3, the dots represent the maximum norms constructed as shown in the proof, while the lines depict the theoretical bounds. Overall, this setup is designed to numerically verify the construction in Theorem 2 by analyzing how the norm of weights decreases as the number of layers increases. In particular, we observe that the maximum norms decrease at the rate predicted by Theorem 2 as the number of layers increases.

## D  Experiment Details

We run each experiment multiple times, but we report the minimum error across runs in our plots. We would like to avoid the influence of optimization error intentionally, because our theoretical results are concerned with the representational capacity of deep linear SSMs, assuming ideal optimization.

As for the computational cost, let us take the example of an $l$-layer SSM of width $m$ compared with a one-layer SSM of width $l(m-1)+1$, which are equivalent in terms of expressivity with the same parameter counts. In one training step, the one-layer SSM makes $2l^2m^2 + o(l^2m^2)$ float calculations (additions and multiplications), while the $l$-layer SSM makes $4lm^2 + o(lm^2)$ float calculations, which is of smaller order than the one-layer SSM. Nevertheless, in practical implementations leveraging parallel computation, the runtime of matrix-vector multiplication does not increase quadratically with matrix size but often remains nearly constant over a wide range of sizes, due to efficient hardware utilization. The one-layer SSM requires one such calculations per timestep while the $l$-layer SSM requires $2l$ such calculations, thus leading to slower calculations in forward pass. As we consider backward pass to have the same order of calculations as forward pass, we have that the one-layer SSM to be faster to train.

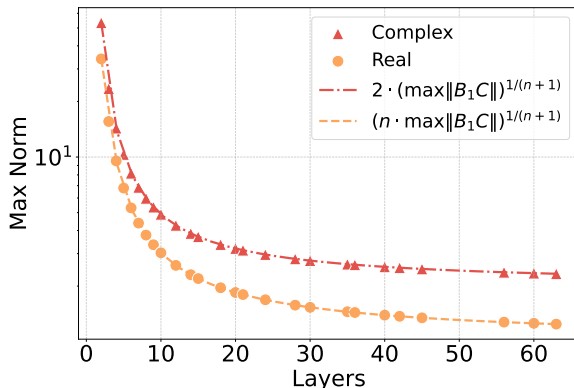

Figure 3: We use a one-layer linear network to represent a deep linear network under reconstruction setting, examining the relationship between the number of layers and the maximum norm of the corresponding parameters in the model.

## D.1 ERROR BARS FOR SECTION 5.1

| Layer | Loss (Mean ± 95%CI) |
|-------|---------------------|
| 1 | $(5.70 \pm 0.29) \times 10^{-2}$ |
| 2 | $(4.77 \pm 0.13) \times 10^{-2}$ |
| 3 | $(7.35 \pm 0.47) \times 10^{-3}$ |
| 5 | $(1.84 \pm 0.68) \times 10^{-4}$ |

Table 3: Loss (Mean ± Standard Deviation) for different layers in learning linear impluse

## D.2 ERROR BARS FOR SECTION 5.2

| Layer | Accuracy (Mean ± 95%CI) |
|-------|-------------------------|
| 1 | $96.06\% \pm 0.22\%$ |
| 2 | $97.28\% \pm 0.13\%$ |
| 3 | $98.33\% \pm 0.11\%$ |
| 4 | $98.48\% \pm 0.12\%$ |
| 5 | $98.51\% \pm 0.14\%$ |

Table 4: Accuracy (Mean ± Standard Deviation) for different layers on MNIST

## D.3 NONLINEAR S4D MODEL ON CIFAR-10 DATASET

We conducted additional experiments using nonlinear S4D models, where each linear SSM block is followed by a feedforward neural network (FFN), on the CIFAR-10 dataset within the Long Range Arena (LRA) benchmark (Tay et al., 2021). The results demonstrate that increasing the model depth, while keeping the total parameter count approximately constant, leads to improved test accuracy. Specifically, models with greater depth and narrower width consistently outperform shallower, wider counterparts. The test accuracies for different model configurations are summarized below:

| Depth | Width | Test Accuracy |
|-------|-------|---------------|
| 4 | 64 | 0.8048 |
| 3 | 86 | 0.8036 |
| 2 | 128 | 0.7978 |
| 1 | 256 | 0.7846 |

Table 5: Test accuracies for different model configurations on CIFAR-10

### D.4 INTUITIVE ILLUSTRATION FOR NONLINEAR S4D MODEL IN SECTION 5.2

The model architecture used in our experiments on the nonlinear S4 model for the MNIST dataset is

$$\text{SSM}_{l,m} \to \text{FFN}$$

where FFN denotes a feedforward network. In this equation, the arrow denotes the sequential composition of linear SSM layer with different depth and width and FFN. An illustrative figure could be described as follows:

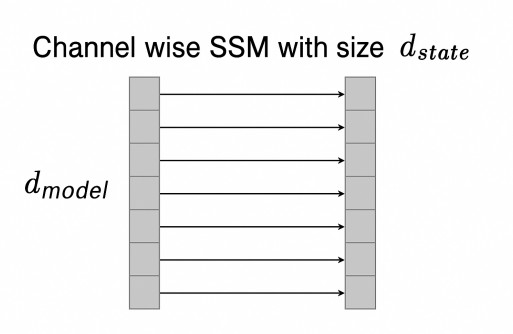

## E LARGE LANGUAGE MODEL USAGE

Large language models were used solely for minor language editing (e.g., polishing and grammar checks). All conceptual development, theoretical results, experimental design, and analyses are the authors' own work.

