# OpenReview forum: "The Effect of Depth on the Expressivity of Deep Linear State-Space Models"
_ICLR.cc/2026/Conference — Submitted to ICLR 2026_

### Official Review · Reviewer_XwKy · 2025-10-15

**Soundness:** 3
**Presentation:** 3
**Contribution:** 3
**Rating:** 6
**Confidence:** 4

**Summary:**

State-space models (SSMs) are a class of sequential models that have recently gained significant interest in the deep learning community due to their strong performance and computational efficiency. While a single SSM layer is remarkably simple, their power emerges from stacking multiple layers and interleaving them with MLPs. This paper provides valuable theoretical insights into this phenomenon by analyzing, in the linear regime (i.e., without MLPs), how depth affects the network's expressive power. The authors demonstrate that while networks of all depths are equivalent given the same number of hidden neurons, deeper networks achieve this expressivity with parameters of lower norms, which facilitates learning.

**Strengths:**

The paper is well-written overall, and the theoretical results formalize intuitive yet non-trivial insights. By addressing a gap in our theoretical understanding of these models, this work makes a valuable contribution to the community.

**Weaknesses:**

The primary weakness lies in the empirical section, which consists mainly of relatively shallow empirical verification of the theoretical results without adding substantial value. While experiments demonstrate that deep linear SSMs are easier to learn, they do not explain why this occurs. It would strengthen the paper to connect these findings more explicitly to the theory, for example, by showing that optimal parameters are closer to initialization or that the loss landscape is better conditioned. Additionally, in the S4D experiment, it remains unclear whether the improved performance stems from easier learning or simply from having more nonlinear layers. As a minor note, the wall-clock time measurements could be removed without diminishing the paper's quality.

**Questions:**

The teacher-student experiment in Figure 3 requires clarification. While it is described as a teacher-student setup (implying a learning experiment), the perfect matching in the plot suggests that what is being reported is the norm from the construction itself rather than learned parameters. I would expect that student over-parameterization would be necessary for successful teacher mimicking, and that the empirical results would exhibit more variance. Could the authors clarify what exactly is done here?

Additionally, here are several relevant papers in the SSM theory space could help better position this work:
- [Saxe et al. (2013)](https://arxiv.org/pdf/1312.6120) - Analysis of deep linear networks
- [Orvieto et al. (2024)](https://arxiv.org/pdf/2307.11888) - Complex vs. real recurrence, particularly demonstrating how complex numbers significantly reduce parameter norms
- [Zucchet & Orvieto (2024)](https://proceedings.neurips.cc/paper_files/paper/2024/file/fbb07254ef01868967dc891ea3fa6c13-Paper-Conference.pdf) - How SSM design affects optimization, with focus on the challenges of learning the A matrix (complementary to this paper)
- [Proca et al. (2025)](https://openreview.net/pdf?id=KGOcrIWYnx) - Learning dynamics of linear SSMs

---

> ### Author Response · Authors · 2025-11-22
> **Reply to weakness (part 1)**
>
> **W1: The primary weakness lies in the empirical section, which consists mainly of relatively shallow empirical verification of the theoretical results without adding substantial value. While experiments demonstrate that deep linear SSMs are easier to learn, they do not explain why this occurs.**
>
> Instead of proposing training heuristics, we are proposing architecture design. We propose to replace the linear layer by multiple linear layers, depending on the difficulty of the target. This is a an initial step to more systematic architecture design. Traditionally, a standard nonlinear S4D block consists of a single linear SSM followed by a nonlinear FFN, which is then stacked to form a multi-layer architecture. In our work, we replace the single-layer linear SSM with a multi-layer linear SSM, since they have the same expressivity without the parameter norm constraints. This leads to a potentially more efficient architecture. First, Theorem 4.1 shows that at constant parameter counts, an $l$-layer linear SSM does not lose expressivity compared to a single-layer SSM, provided the total number of parameters is sufficient. Second, a multi-layer SSM can approximate the same targets with smaller parameter norms than a single-layer SSM. From Theorem 4.3, the optimal choice of $l$ depends on the norm required by a single-layer SSM to approximate the target: the larger this norm, the more layers are beneficial. Third, while increasing the number of layers can help reduce parameter norms, it may also increase computation time due to computation parallelization. Thus, there is a trade-off: shallow networks are computationally efficient but may suffer from large parameter norms, whereas deeper networks can reduce parameter norms but incur higher computational costs. Our results provide guidance for selecting $l$ to balance parameter norm and computational efficiency under the equivalent expressivity.
>
> Our empirical section was designed to validate the constructive theory from Theorem 1 to Theorem 3 rather than to propose new training heuristics. We keep the parameter count fixed by reducing the width $m$ as depth $l$ increases by enforcing $l(m-1) + 1$ be constant. A deeper network therefore uses smaller matrices per layer. The total flop count scales roughly like $O(lm^{2})$ which does not grow linearly with $l$ once $m$ is shrinking. This analysis is already explained in Appendix C.
>
> In particular, we add a new experiment explaining why parameter norm control is important. We train linear SSMs with different depth and width but we  keep the same parameter counts to learn the impulse memory functions defined as equation (15) at positions $t=10$ and $t=50$ and the results are shown in the table below.
>
> | Architecture        | steps (190s), $t=10$ | loss, $t=10$                | steps (190s), $t=50$ | loss, $t=50$                    |
> |---------------------|----------------------|-----------------------------|----------------------|---------------------------------|
> | 1 layer, width 31   | 3328                 | **$2.1\times 10^{-5}$**     | 3328                 | $1.3\times 10^{-3}$             |
> | 2 layers, width 16  | 2195                 | $2.3\times 10^{-5}$         | 2199                 | $5.6\times 10^{-4}$             |
> | 3 layers, width 11  | 1603                 | $3.4\times 10^{-5}$         | 1613                 | **$5.3\times 10^{-4}$**         |
>
> The impulse memory function with $t=10$ is a ''small-norm target'' whereas that with $t=50$ is a relatively ''large-norm target''.
> By this we mean that to learn the target for $t=50$ at fixed model sizes, it is known that one requires larger norms in the trainable parameters to reach a similar approximation error ([1]).  All models are trained for approximately the same wall-clock time ($\approx 190$s), so deeper models see fewer optimization steps. For the impulse memory function target at $t=10$, the shallow 1-layer SSM converges fastest and attains the lowest loss. For the impulse memory function target at $t=50$ which requires larger parameter norms to reach a similar approximation error, the deepest 3-layer SSM achieves the best loss, even though it runs slower per step.
> This is consistent with our Theorem 2: depth could help reduce the norm. A deep SSM can realize a large effective kernel norm by factorizing it into a product of smaller per-layer norms, whereas a shallowm SSM cannot, making the large-norm target harder to approximate under norm constraints.
>
> **Reference**
>
> [1]: Fusheng Liu and Qianxiao Li. Autocorrelation matters: Understanding the role of initialization schemes for state space models. In The Thirteenth International Conference on Learning Representations, 2025.

---

> ### Author Response · Authors · 2025-11-22
> **Reply to weakness (part 2)**
>
> **W2: It would strengthen the paper to connect these findings more explicitly to the theory, for example, by showing that optimal parameters are closer to initialization or that the loss landscape is better conditioned.**
>
> It is a very interesting future work to discuss optimization dynamics and landscape, but in this paper we are focusing strictly on expressivity in the presence of parameter norm constraints. The relevant theoretical result for section 5.2 is Theorem 4.2, which shows that the norm of the weights in deep linear SSMs decreases exponentially as the number of layers increases. In the third experiment, due to the choice of optimization algorithms (e.g. weight decay) or initilization schemes (small norm initialization), the parameter norm of each layer will be effectively upper bounded by some constant, denoted as $B$. Then according to Theorem 4.2, the max parameter norm of the equivalent one layer SSM that can be expressed by such $l$ layer SSM would be upper bounded by $O(B^{l/2})$, which is exponential to $l$. Each point of the yellow line in the left panel of Figure 2 is the max norm of weights equivalent to one layer nonlinear S4D across different layers while keeping the same parameter counts. We can see that the max norm of weights equivalent to one layer S4D increases exponentially with increasing layers. While Theorem 4.2 is established in a linear setting, we observe a qualitatively similar trend in Figure 3 for the nonlinear S4D model trained on MNIST and Cifar-10: increasing depth leads to a substantial reduction in model norm. Due to the nonlinearity of S4D, this experiment does not offer an exact quantitative match to the theory, but it suggests that the norm exponential reduction phenomenon may generalize beyond linear models.
>
> **W3: Additionally, in the S4D experiment, it remains unclear whether the improved performance stems from easier learning or simply from having more nonlinear layers. As a minor note, the wall-clock time measurements could be removed without diminishing the paper's quality.**
>
> Thanks for your suggestions. In our S4D setup, we do not add more nonlinear layers when we increase SSM depth. We stack linear SSM blocks with no internal nonlinearity and append a single, identical FFN across all configurations. Only depth and width of the linear SSM stack are varied to keep the SSM parameter budget fixed. Thus, the number of nonlinear layers is held constant and the performance in Figure 2 can be attributed to the deeper SSM factorization under the same budget, which is consistent with norm‑reduction mechanism as shown in Theorem 2.
>
> **W4: The teacher-student experiment in Figure 3 requires clarificationWhile it is described as a teacher-student setup (implying a learning experiment), the perfect matching in the plot suggests that what is being reported is the norm from the construction itself rather than learned parameters. I would expect that student over-parameterization would be necessary for successful teacher mimicking, and that the empirical results would exhibit more variance. Could the authors clarify what exactly is done here?**
>
> We agree that the phrase “teacher–student” is confusing and we will revise it into a ``reconstruction experiment''. Over‑parameterization is often helpful for optimization, but it is not necessary for representability in our linear setting. Our Theorem 1 shows width–depth equivalence without norm constraints and our Theorem 2 then shows how to factor the same kernel under norm bounds at the same order of parameters. In Appendix B (Figure 3), we do not rely on gradient‑based training to approximate the teacher. Instead, for each depth $l$, we start from a fixed one‑layer teacher SSM with width 2521 and apply the explicit constructive mapping given in the proof of Theorem 2 to obtain the parameters $A_{1},...,A_{l},B_{1},...,B_{l},C$ of an $l$-layer student SSM with width $m$ satisfying $2521=l(m-1)+1$. By construction, the student network realizes exactly the same convolutional kernel $\rho(t)$ as the teacher for all $t$. Thus, there is no optimization or convergence issue in this particular experiment and the student networks are exactly equivalent to the teacher by the explicit formula from Theorem 2. We used the term “teacher–student setup” only to conceptually distinguish the wide one‑layer model (teacher) from the constructed deep models (students) by keeping the same parameter counts. Figure 3 is a direct illustration of representability under these theorems and it is not an SGD learning curve. It is a deterministic verification of the constructive factorization in Theorem 2. Specifically, given a one-layer linear SSM with width $l(m-1)+1$, we apply the construction from Theorem 2 to reconstruct an equivalent model with $l$ layers and width $m$.

---

> ### Comment · Reviewer_XwKy · 2025-11-22
>
> I thank the authors for the rebuttal and for the clarification about Figure 2. I believe that the response to my comments and to the ones of other reviewers will improve the clarity and impact of the paper; I encourage the authors to integrate them in the next version of the paper (as well as a more detailed discussion of existing work, cf. my review).
>
> I have read the other reviews, in particular the negative ones of `TrMR` and `8W94`, and they do not change my evaluation of the paper: `TrMR` does not raise any fundamental limitation of the paper, `8W94`'s expectation are, in my opinion, way beyond what is reasonable to expect from a SSM theory paper.
>
> As a result, I strongly believe that the paper should be accepted, and have updated my score from 6 to 8.

---

### Official Review · Reviewer_8W94 · 2025-10-25

**Soundness:** 3
**Presentation:** 3
**Contribution:** 1
**Rating:** 2
**Confidence:** 3

**Summary:**

This paper explores a tradeoff between width and depth in a linear variant of common deep state space models.  A theoretical analysis is presented by considering alternative representations of linear SSMs.  Theoretical results are backed by some empirical results.

**Strengths:**

1. Study of practical trade-offs within the design of deep SSMs is (imo) chronically lacking in the literature.  This paper is a step towards that.
2. The paper is relatively well written, and is relatively easy to follow.
3. The experiments do support some of the claims made.

**Weaknesses:**

# Major Weaknesses

My major block with this paper is what the purpose of the paper is and the level of significance it achieves:

1. In practice, deep SSMs use always non-linearities.  Therefore, I will always struggle, on a fairly fundamental level, to be too enthusiastic about work that studies a theoretical construct.  This is not a comment so much on the quality of the work, but the significance.*  This may be acceptable for learning-theory specific conferences or workshops, but these types of work, in my _opinion_, rarely meet the level of significance I see in other papers.
2. Building on this, the core of the study is that coupled linear systems can be represented as one larger linear system, and then performs an analysis of the parameter norms/predictive accuracies across different depths/widths of systems (and their corresponding implied models).  This is somewhat interesting, but I feel like it is not that impactful as a  study, nor does it introduce any methodological insights, again, harming the significance of the work.
3. There is little-to-no meaningful discussion of how this discovery helps us build better (even linear, let alone non-linear) deep SSMs.  When should I use a deep network vs a shallow network?  The analysis doesn't actually link this to help design better networks, it just explains some phenomena that you might see while already training networks.
4. Following on from this again, there is no justification of why I should worry about the norm of my parameters?  In my understanding, norm constraints (hard or soft) are useful for limiting the expressiveness of very large networks on limited data (i.e. regularization).  But this work looks at networks with fixed expressivity (at least in the unconstrained norm case), so regularization isn't relevant.
5. I think there is some admission of this in the conclusion and future work sections ("Understanding this trade-off more thoroughly in nonlinear or real-world SSM variants remains an exciting avenue for future exploration.").  The points raised in these sections, to me, feel like the payoff for the theoretical examination that is presented, i.e. this is a great preliminary and setup for more wide-ranging experiments and further theoretical innovation.


I will also state that I really struggle to understand the experiment in Section 5.2:  I don't understand the experimental setup, the purpose, or the findings.  For instance: what does the non-linear SSM experiment in Table 1 show (we know deeper networks are better all the way back to VGG?)?  You claim models have the same expressivity, but one has better performance, implying they have different expressivity?  What is the significance of the FFN in this experiment?  Why is it needed?  What happens if it is removed?  I invite the authors to either write a comment re-affirming the experiment and findings, or, re-write the section in the paper (I believe you're allowed to do this at ICLR?) and I commit to re-evaluating the revised manuscript.  Right now, however, I simply take nothing meaningful from this experiment, which is a shame.

# Minor Weaknesses:
1. Can you include the norm of the parameters for the multi-layer networks on the diagrams?  Your initial claim is that the norm of parameters is lower in deeper networks.  As far as I can tell, this isn't actually quantitatively reported anywhere.
2. Why is the slope of the wall clock time not equal to 1?  Shouldn't a five-layer network be five times more expensive than a one-layer network?
3. Numerous citations are incorrect, e.g. S4 is from Gu not Smith.
4. Smith isn't one-dimensional (line 159).
5. Do the norms also have error bars?
6. A 2x increase in wall clock time for a dramatic reduction in error seems (at least in broad strokes) acceptable to me.
7. Are there differences in the memory consumption for different models?  Both in terms of on-disk memory footprint (presumably not in the number of parameters are roughly consistent) and GPU-memory consumption?  Memory is often as important as factor as runtime.
8. This is personal preference, but I like it when papers with lots of theorems and lemma blocks follow each with a "Remarks" block, which expands the high-level discussion or proof sketch.
9. I'm less sure on this one, but there is a wealth of work on learning and expressivity in linear networks.  It would be a nice touchpoint if you viewed the SSM as one big linear FFN (which you can do for fixed-length inputs using Toeplitz matrices).  Right now the work is presented as living in a little bit of a vacuum, which I think is a missed opportunity.

# Conclusion:

Ultimately, I am left struggling with what the purpose or key take-away from this work is, and so I am left underwhelmed by this paper and struggling to find reasons to argue for its inclusion.  For now I will reduce my confidence from 4 to 3, because I am cognizant that I might have simply missed the point of this paper.  I am also open to re-evaluating my score if the authors can allay my concerns on the significance.

I think this work is undoubtedly a start on a path to bigger and more practicable insights, but I simply do not think it is there yet.  I strongly encourage the authors to continue working on this, and look to develop more practicable insights on models that are used in practice (e.g. Mamba).  Then this work could be incredibly impactful.  Good luck.

(*If the authors can show works that use deep linear SSMs and are competitive, or if they benchmark them themselves, then this could be assuaged, but this is not present as it stands.)

**Questions:**

I invite the authors to respond to the criticisms I raise above in weaknesses, specifically justifying the broad significance and impact of this work.

---

> ### Author Response · Authors · 2025-11-22
> **Reply to weakness (part 1)**
>
> **W1: In practice, deep SSMs use always non-linearities. Therefore, I will always struggle, on a fairly fundamental level, to be too enthusiastic about work that studies a theoretical construct. This is not a comment so much on the quality of the work, but the significance. This may be acceptable for learning-theory specific conferences or workshops, but these types of work, in my opinion, rarely meet the level of significance I see in other papers.**
>
> We understand the concern, and we agree that practical SSMs (S4, Mamba, etc.) are nonlinear. Our goal in this paper is to understand how depth affects the expressivity of deep SSMs and the tradeoff between depth and width under norm constraints. To the best of our knowledge, this is the first rigorous analysis of (norm constrained) capacity of deep linear SSMs, offering theoretical insights into their norm weights and representational abilities as depth increases. Our theoretical results are valuable by revealing the role of depth and width under the parameters norm in deep linear SSMs. It is potentially useful to extend our theoretical framework to designing computational efficient architecture.
>
> Linear SSMs are not meant to be realistic standalone models and they are the linear backbone sitting inside modern SSM layers. Our results therefore target the core mechanism that nonlinear SSMs build on. This is analogous to the extensive literature on deep linear networks that has influenced our understanding of implicit bias, norm dynamics and optimization in nonlinear networks. Empirical evidence that the same qualitative trade‑off appears in a nonlinear S4D architecture on sequential MNIST and CIFAR‑10 under fixed parameter budgets (Section 5.2).
>
> **W2: Building on this, the core of the study is that coupled linear systems can be represented as one larger linear system, and then performs an analysis of the parameter norms/predictive accuracies across different depths/widths of systems (and their corresponding implied models). This is somewhat interesting, but I feel like it is not that impactful as a study, nor does it introduce any methodological insights, again, harming the significance of the work.**
>
> 'Coupled linear systems can be represented as
> one larger linear system' is only the starting point. Firstly, we do not just note that an $l$-layer SSM can be written as a single layer. As shown in Theorem 1, we prove that the width and depth are equivalent in expressivity without norm constraints and we constructively prove that the bound $l(m-1)+1$ is tight. This shows that the multilayer coupled linear system can be expressed optimally as a single layer linear system, so that there is no benefit to parameter counts whether or not one uses the multilevel or the single level one. This is an important point: without further norm constraints, there is no advantage to use deep linear SSMs. Secondly, we give a constructive factorization that transforms a shallow, high‑norm SSM into a deeper, low‑norm SSM under the same parameter count and provide explicit upper bounds on the required norms. This is a new result specific to the SSM setting and norm‑constrained expressivity. Thirdly, given a one-layer linear SSM with width $K$ and a per‑layer norm budget $c_{2}$, we derive an explicit upper bound on the depth needed to realize it with per‑layer norm at most $c_{2}$. This yields a concrete design rule: for a given desired memory complexity and norm constraint, we can estimate how many layers we need. We will emphasize these aspects more prominently and explicitly highlight the constructive nature of our proofs as a methodological contribution, which provide a method for factoring shallow SSMs into deeper SSMs under norm constraints.

---

> ### Author Response · Authors · 2025-11-22
> **Reply to weakness (part 2)**
>
> **W3: There is little-to-no meaningful discussion of how this discovery helps us build better (even linear, let alone non-linear) deep SSMs. When should I use a deep network vs a shallow network? The analysis doesn't actually link this to help design better networks, it just explains some phenomena that you might see while already training networks.**
>
> We appreciate this comment. Our results suggest the following concrete design principles. Firstly, Theorem 2 shows that if we need a setting where per‑layer norms are constrained by stability, regularization or optimizer behavior (weight decay in Adam optimizer) and we need to learn a target function which requires larger norms in the trainable parameters to reach a similar approximation error (e.g., impulse at far distance defined in equation (15)), we should prefer deeper SSMs for the same parameter budget. Secondly, Theorem 1 shows that if norms are unconstrained and data are plentiful, our theory suggests there is no intrinsic reason to favor depth on expressivity grounds. Thirdly, a standard nonlinear S4D block consists of a single linear SSM followed by a nonlinear FFN, which is then stacked to form a multi-layer architecture. In our work, we replace the single-layer linear SSM with a multi-layer linear SSM, since they have the same expressivity without the parameter norm constraints. From the blue line in the left panel of Figure 2, we could see that increasing the number of layers, while reducing the hidden dimension to maintain the same expressivity, can lead to improved model performance. Last but not least,  while increasing the number of layers can help reduce parameter norms, it may also increase computation time due to computation parallelization. Thus, there is a trade-off: shallow networks are computationally efficient but may suffer from large parameter norms, whereas deeper networks can reduce parameter norms but incur higher computational costs. Our results provide guidance for selecting to balance parameter norm and computational efficiency under the equivalent expressivity.
>
> **W4: I think there is some admission of this in the conclusion and future work sections ("Understanding this trade-off more thoroughly in nonlinear or real-world SSM variants remains an exciting avenue for future exploration."). The points raised in these sections, to me, feel like the payoff for the theoretical examination that is presented, i.e. this is a great preliminary and setup for more wide-ranging experiments and further theoretical innovation.**
>
> We appreciate this perspective and agree that extending these ideas to nonlinear/state‑selective models (e.g., Mamba) is an important next step. At the same time, we want to emphasize that the current paper already provides: 1. The first rigorous capacity analysis of deep linear SSMs under norm constraints, with tight bounds and constructive factorization. 2. A clear separation between the norm‑unconstrained regime (depth vs width equivalent) and the norm‑constrained regime (depth strictly more powerful). 3. Empirical evidence linking these theoretical phenomena to nonlinear S4D models on real tasks. We will emphasize more clearly our contributions in this regard in a revision.
>
> **W5: What is the significance of the FFN in this experiment? Why is it needed? What happens if it is removed? I invite the authors to either write a comment re-affirming the experiment and findings, or, re-write the section in the paper (I believe you're allowed to do this at ICLR?) and I commit to re-evaluating the revised manuscript. Right now, however, I simply take nothing meaningful from this experiment, which is a shame.**
>
> Sequential MNIST and CIFAR‑10 are classification tasks that require nonlinearity. A purely linear model over the raw sequence or convolutional kernel would be unable to reach competitive accuracy. We therefore follow the standard S4D architecture: linear SSM layers followed by a nonlinear FFN. To isolate the effect of depth in the SSM, we fix the FFN to be identical across all configurations and only change the SSM depth and width while keeping the same parameter counts. Due to the nonlinearity of S4D, this experiment does not offer an exact quantitative match to the theory, but it suggests that the norm exponential reduction phenomenon may generalize beyond linear models which is the initial point of adding FFN to linear SSM layers.

---

> ### Author Response · Authors · 2025-11-22
> **Reply to weakness (part 3)**
>
> **W6: Following on from this again, there is no justification of why I should worry about the norm of my parameters? In my understanding, norm constraints (hard or soft) are useful for limiting the expressiveness of very large networks on limited data (i.e. regularization). But this work looks at networks with fixed expressivity (at least in the unconstrained norm case), so regularization isn't relevant.**
>
>
> As mentioned from line 161 to line 167 in the paper, state-space models may suffer from large parameter norms when learning specific memory functions, which may lead to training instability and negatively impact model performance [1]. Theoretically, [2] proves that the parameter norm of SSMs would be at least exponentially increasing with respect to width if we would like to keep the same budgets to approximate specific target relationships. As you noted, unconstrained expressivity is fixed once parameter count is fixed. Our key point as shown in Theorem 2 is that in the norm‑constrained regime where many applications implicitly adopt (e.g., weight decay during training, small initialization), depth and width are no longer equivalent and depth could make more expressive convolutional kernels without blowing up norms. Even without explicit hard constraints, weight decay, spectral normalization and optimizer dynamics implicitly bias towards low‑norm solutions, which are the subset of convolutional kernels realizable under the norm constraints. Our Theorem 2 say that this subset grows with depth, which means that deeper SSMs can reach kernels that shallow ones cannot access under norm constraints with the same parameter counts.
>
> In particular, we add a new experiment explaining why parameter norm control is important. We train linear SSMs with different depth and width but we  keep the same parameter counts to learn the impulse memory functions defined as equation (15) at positions $t=10$ and $t=50$ and the results are shown in the table below.
>
> | Architecture        | steps (190s), $t=10$ | loss, $t=10$                | steps (190s), $t=50$ | loss, $t=50$                    |
> |---------------------|----------------------|-----------------------------|----------------------|---------------------------------|
> | 1 layer, width 31   | 3328                 | **$2.1\times 10^{-5}$**     | 3328                 | $1.3\times 10^{-3}$             |
> | 2 layers, width 16  | 2195                 | $2.3\times 10^{-5}$         | 2199                 | $5.6\times 10^{-4}$             |
> | 3 layers, width 11  | 1603                 | $3.4\times 10^{-5}$         | 1613                 | **$5.3\times 10^{-4}$**         |
>
> The impulse memory function with $t=10$ is a ''small-norm target'' whereas that with $t=50$ is a relatively ''large-norm target''.
> By this we mean that to learn the target for $t=50$ at fixed model sizes, it is known that one requires larger norms in the trainable parameters to reach a similar approximation error ([1]). All models are trained for approximately the same wall-clock time ($\approx 190$s), so deeper models see fewer optimization steps. For the impulse memory function target at $t=10$, the shallow 1-layer SSM converges fastest and attains the lowest loss. For the impulse memory function target at $t=50$ which requires larger parameter norms to reach a similar approximation error, the deepest 3-layer SSM achieves the best loss, even though it runs slower per step.
> This is consistent with our Theorem 2: depth could help reduce the norm. A deep SSM can realize a large effective kernel norm by factorizing it into a product of smaller per-layer norms, whereas a shallowm SSM cannot, making the large-norm target harder to approximate under norm constraints.
>
> **Reference**
>
> [1] Haotian Jiang, Zeyu Bao, Shida Wang, and Qianxiao Li. Numerical investigation of sequence modeling theory using controllable memory functions. arXiv preprint arXiv:2506.05678, 2025.
> [2] Fusheng Liu and Qianxiao Li. Autocorrelation matters: Understanding the role of initialization schemes for state space models. In The Thirteenth International Conference on Learning Representations, 2025.

---

> ### Author Response · Authors · 2025-11-22
> **Reply to weakness (part 4)**
>
> **W7: I will also state that I really struggle to understand the experiment in Section 5.2: I don't understand the experimental setup, the purpose, or the findings. For instance: what does the non-linear SSM experiment in Table 1 show (we know deeper networks are better all the way back to VGG?)? You claim models have the same expressivity, but one has better performance, implying they have different expressivity?**
>
> For the experiment setup, we start with the S4D architecture. We consider an SSM backbone composed of $l$ linear SSM layers stacked sequentially, with no internal nonlinearity between them. We adjust the state dimension $d_{state}$ as we vary $l$ to keep the parameter counts of SSM approximately constant and use a fixed FFN across configurations.
>
> The relevant theoretical result for section 5.2 is Theorem 4.2, which shows that the norm of the weights in deep linear SSMs decreases exponentially as the number of layers increases. Each point of the yellow line in the left panel of figure 2 is the max norm of weights equivalent to one layer nonlinear S4D across different layers while keeping the same parameter counts. We can see that the max norm of weights equivalent to one layer S4D increases exponentially with increasing layers. While Theorem 4.2 is established in a linear setting, we observe a qualitatively similar trend in Figure 2 for the nonlinear S4D model trained on real task: increasing depth leads to a substantial reduction in model norm. Due to the nonlinearity of S4D, this experiment does not offer an exact quantitative match to the theory, but it suggests that the norm exponential reduction phenomenon may generalize beyond linear models.
>
> What we mean by same expressivity here is that under fixed parameter counts and no norm constraints, depth and width are equivalent and their trades in linear SSMs do not change the maximum expressivity. In practice, however, training is implicitly norm-constrained under nonlinear architectures (e.g. weight decay, small initialization). Thus, while the theoretical unconstrained expressivity is comparable across architectures, their effective expressivity under norm and optimization bias differs.
>
> **W8: Can you include the norm of the parameters for the multi-layer networks on the diagrams? Your initial claim is that the norm of parameters is lower in deeper networks. As far as I can tell, this isn't actually quantitatively reported anywhere.**
>
> hank you for this suggestion. In Figures 1 and 2, we currently show the norm of the equivalent one‑layer SSM computed via Lemma 2. We agree it would be informative to also report the max parameter norms of the equivalent one-layer.  For figure 1, we report norm of parameters as follows:
>
> For Figure 1, we report the max parameter norm as follows:
>
> | Depth | Width | Max parameter norm |
> |-------|-------|--------------------|
> | 1     | 31    | 0.294940           |
> | 2     | 16    | 1.992066           |
> | 3     | 11    | 2.138185           |
> | 5     | 7     | 3.411252           |
>
> For Figure 2, we report the max parameter norm as follows:
>
> | Depth | Width | Max parameter norm |
> |-------|-------|--------------------|
> | 1     | 256   | 24.318537          |
> | 2     | 128   | 66.96496           |
> | 3     | 86    | 260.85623          |
> | 4     | 65    | 560.29297          |
> | 5     | 52    | 4021.407           |
>
> **W9: Why is the slope of the wall clock time not equal to 1? Shouldn't a five-layer network be five times more expensive than a one-layer network?**
>
> Width changes with depth. To keep the parameter count fixed, we reduce the width $m$ as depth l increases by enforcing $l(m-1) + 1$ be constant. A deeper network therefore uses smaller matrices per layer. The total flop count scales roughly like $O(lm^{2})$ which does not grow linearly with $l$ once $m$ is shrinking. This analysis is already explained in Appendix C. Furthermore, on modern hardware, the runtime of matrix–vector operations is not strictly proportional to the number of layers because larger matrices are often more efficient than many small ones and multiple layers can sometimes be fused or overlapped.

---

> ### Author Response · Authors · 2025-11-22
> **Reply to weakness (part 5)**
>
> **W10: Are there differences in the memory consumption for different models? Both in terms of on-disk memory footprint (presumably not in the number of parameters are roughly consistent) and GPU-memory consumption? Memory is often as important as factor as runtime.**
>
> We did not report memory measurements explicitly, but we will add this to discussion. Under our design $l(m-1)+1$ be constant, the total number of parameters in the SSM portion is approximately constant across depths. Thus the on‑disk model size remains essentially unchanged. During training, memory is dominated by parameters (constant under fixed parameter budget) and activations. For sequence length $T$, depth $l$, and width $m$, activation memory is roughly $O(T l m)$. Since we reduce $m$ as $l$ increases while keeping $l(m - 1)$ be constant, the product $lm$ stays roughly constant as well. Therefore, both parameter and activation memory should be similar across depths under our scaling.
>
> We train linear SSMs to learn impulse function at $\alpha=10$. We run two epochs for deep linear SSM as defined in (1) and record checkpoint size on disk (MB) and peak GPU memory allocated \& reserved (MB).
> | Depth $L$ | Width $m$ | Checkpoint size (MB) | Peak GPU memory allocated (MB) |
> |-----------|-----------|----------------------|---------------------------------|
> | 1         | 301       | 0.01                 | 35.96                           |
> | 2         | 151       | 0.18                 | 36.62                           |
> | 3         | 101       | 0.16                 | 36.41                           |
>
> **W11: I'm less sure on this one, but there is a wealth of work on learning and expressivity in linear networks. It would be a nice touchpoint if you viewed the SSM as one big linear FFN (which you can do for fixed-length inputs using Toeplitz matrices). Right now the work is presented as living in a little bit of a vacuum, which I think is a missed opportunity.**
>
> We agree this is a valuable connection and already cite some works on deep linear networks. Some studies focus on the training dynamics of deep linear networks from an optimization perspective [1]\&[2], whereas we focus on representation with norm constraints. Especially, [2] analyzes the benefits of width in deep linear networks, showing that deep and wide linear neural networks converge to a global minimum with polynomial running time while deep and narrow linear networks converge with
> exponential running time. In this paper, we consider deep linear state-space models (SSMs) which apply to sequence modeling. This fundamental difference leads to distinct findings in investigating the two models. However, relatively little work has been devoted to understanding how depth affects the expressivity of deep SSMs. We adopt a simple formulation that allows us to characterize important differences between multi-layer SSM and one-layer SSM from depth and width tradeoff perspective.
>
> **Reference:**
>
> [1]: Govind Menon. The geometry of the deep linear network. arXiv preprint arXiv:2411.09004, 2024.
>
> [2]: Simon Du and Wei Hu. Width provably matters in optimization for deep linear neural networks. In
> International Conference on Machine Learning, pp. 1655–1664. PMLR, 2019.

---

> ### Comment · Reviewer_8W94 · 2025-11-22
> **Raising with Caveats**
>
> To the authors,
>
> Thank you for your detailed response.  Unfortunately, I am not particularly swayed by any of the arguments put forth.  For me, the major blocker is still that this does not apply to practical SSM architectures, and this fundamentally caps the magnitude of the contribution.  This is borne out in how, even after the author comments, I do not think there are any particularly meaningful findings in this paper, e.g.:
> >It is potentially useful to extend our theoretical framework to designing computational efficient architecture.
>
> Deferring extending theory to, potentially, be useful to the thing that motivated deep SSMs in the first place is problematic to me.  This is a quote from the rebuttal, but the paper itself is littered with unsubstantiated and unexplored claims that this work may result in useful follow-ups down the line.  I appreciate the comments of `XwKy `, but I simply do not agree that claiming "theory" permits totally divorcing ones work from what is practical or used in practice.  There are lots of theoretical SSM works out there that study linear SSMs and their interplay with non-linear models, instead of again totally deferring and conjecturing `our analysis is confined
> to the linear setting; although we expect many conclusions to extend to nonlinear models`.
>
> I am still steadfast in my belief that this work does not rise to the level of significance that I see in other ICLR papers.  I would give this a 3 if there were such an option.  However, I am cognizant that my criticism is based on my _opinion_ as opposed to factual or scientific inaccuracies in the paper, and that others (e.g. `XwKy`) may indeed find this work insightful.  The precise wording of the rating 4 therefore seems more appropriate.  I will therefore raise my rating to 4, but make it clear that, if pressed, I am still against this paper being accepted.
>
> Good luck,
>
> Reviewer 8W94

---

### Official Review · Reviewer_CCUp · 2025-10-31

**Soundness:** 3
**Presentation:** 2
**Contribution:** 3
**Rating:** 6
**Confidence:** 3

**Summary:**

The paper investigates the expressivity of deep linear state-space models with respect to bounded parameter norms. The authors show that in the absence of parameter norm bounds, wide linear SSMs can be equivalently represented by deeper, shallow SSMs. Furthermore, it is shown that a wide linear SSM can be represented using a deep linear SSM with smaller norm bounds. The theoretical results are supported with numerical experiments.

**Strengths:**

The authors provide insights into the expressivity of linear SSMs, providing proofs that wide SSMs can be equivalently represented by deep SSMs. The fact that deep SSMs with smaller norm constraints can represent wide SSMs with larger norm constraints is insightful. Similar trends are shown to exist for nonlinear models. The reviewer believes this is an interesting contribution furthering the understanding of SSMs.

**Weaknesses:**

More effort could be spent improving the presentation and flow of the paper. In Section 3.1 for example, the hidden state of any given layer at time step 0, i.e., $h_l(0)$ is not initialized and the writing could be improved.
The theoretical statements are sometimes vague and imprecise. Overall the paper is tough to follow from a theoretical point of view due to a lack of mathematical rigor. The constant $c_1>0$ is not used in the statement of Theorem 1. In Theorem 2 and Corollary 1, I assume the bound holds for an l-layer SSM which is equivalent to the considered one-layer linear SSM, which is not clearly stated. Lemma 2 and its exhibition was confusing to the reviewer. What is meant by output coefficients? What is meant by output expansion? Please refer to an equation if new terminology is used. What is $\xi$? It was not defined before. The authors state that a detailed proof can be found in the appendix, but the proof simply refers to Corollary 2, which refers to Lemma 1, none of which contain any $\xi$ variable. Simply referring to other proofs does not consist of a detailed proof.

**Questions:**

For the numerical validation of Theorem 2, a teacher-student setup is used to analyse the norm constraint relationship. Do the student networks converge to the exact solution in this example? Is it possible to derive a system of equations to numerically find the exact weights such that the networks are equivalent similar to the constructive example provided in Section 4.2?

---

> ### Author Response · Authors · 2025-11-22
> **Reply to weakness (part 1)**
>
> **W1: In Section 3.1 for example, the hidden state of any given layer at time step 0, i.e., $h_{l}(0)$ is not initialized and the writing could be improved. The theoretical statements are sometimes vague and imprecise.**
>
> Thank you for suggestions. In all our theoretical results, we start from $t=0$ and implicitly assume the standard zero‑state initialization for SSMs, i.e. $h_{l}(-1)=0\quad\forall l=1,...,L$. We shall revise this in the new paper version.
>
> **W2: Overall the paper is tough to follow from a theoretical point of view due to a lack of mathematical rigor. The constant $c_{1}>0$ is not used in the statement of Theorem 1. In Theorem 2 and Corollary 1, I assume the bound holds for an l-layer SSM which is equivalent to the considered one-layer linear SSM, which is not clearly stated.**
>
> We appreciate the detailed feedback and agree that there are places where the statements can be made more precise. The constant $c_{1}>0$ in the current statement of Theorem 1 is unnecessary because Theorem 1 is about the unconstrained hypothesis spaces $H_{\infty,l}^{m}$ and does not use $c_{1}$.
>
> The intended meaning of Theorem 2 is that for every convolutional kernel $\rho$, that can be realized by a one‑layer linear SSM with width $l(m-1)+1$ and parameter norm at most $c_{1}$, there exists an $l$-layer SSM with width $m$ and norm bound $c_{2}$ that realizes the same kernel $\rho$. Moreover, one can always choose such an $l$-layer representation with $c_{2}\le2c_{1}^{\frac{2}{l+1}}$. This is exactly what the formal statement (5) is intended to express, but we agree the explanation below the Theorem 2 did not explicitly say “the $l$-layer SSM is equivalent to the given one‑layer SSM in terms of the convolutional kernel.” We shall revise in the new version.
>
> **W3: Lemma 2 and its exhibition was confusing to the reviewer.**
>
> In Lemma 2, the definition of $\xi$ is as follows: A deep linear SSM (Equation (1)) is equivalent to a one‑layer SSM whose kernel is a linear combination of exponentials corresponding to the eigenvalues $\lambda_{i,j}$ of the matrices $A_{i}$. More concretely, under the assumptions of Lemma 2, the output of the deep SSM can be written as $y(t)=\sum_{\tilde{l}=1}^{l}\sum_{\tilde{m}=1}^{m}\xi_{\tilde{l},\tilde{m}}\lambda_{\tilde{l},\tilde{m}}^{t}$.
> This is the definition of the $\xi_{\tilde{l},\tilde{m}}$.
>
> Lemma 2 directly follows from Corollary 2. For completeness, we give the proof below. By Corollary 2 (Equations (32)–(33)), the convolution kernel of the $l$-layer deep linear SSM is $\rho(t)$ where for diagonal $A_i$, we set $\alpha(i,j)=(A_i)_{jj}$.
>
> We introduce the indices $t_{1}=j_{l},...,t_{l}=j_{1}$ and substitute $F_{t}$ in Corollary 2 using Definition 1 could directly get the proof of Lemma 2. We shall improve Lemma 2 formulation and give detailed proof in the revised version.
>
> **W4: For the numerical validation of Theorem 2, a teacher-student setup is used to analyze the norm constraint relationship. Do the student networks converge to the exact solution in this example? Is it possible to derive a system of equations to numerically find the exact weights such that the networks are equivalent similar to the constructive example provided in Section 4.2?**
>
> In Appendix B (Figure 3), we do not rely on gradient‑based training to approximate the teacher. Instead, for each depth $l$, we start from a fixed one‑layer teacher SSM with width 2521 and apply the explicit constructive mapping given in the proof of Theorem 2 to obtain the parameters $A_{1},...,A_{l},B_{1},...,B_{l},C$ of an $l$-layer student SSM with width $m$ satisfying $2521=l(m-1)+1$. By construction, the student network realizes exactly the same convolutional kernel $\rho(t)$ as the teacher for all $t$. Thus, there is no optimization or convergence issue in this particular experiment and the student networks are exactly equivalent to the teacher by the explicit formula from Theorem 2. We used the term “teacher–student setup” only to conceptually distinguish the wide one‑layer model (teacher) from the constructed deep models (students) by keeping the same parameter counts. In section 5, we do train multi-layer linear SSMs with different width and depth by keeping the same parameter counts based on results from Theorem 1.

---

### Official Review · Reviewer_TtMR · 2025-11-06

**Soundness:** 2
**Presentation:** 2
**Contribution:** 3
**Rating:** 4
**Confidence:** 2

**Summary:**

The paper aims to understand how depth influences the expressive capacity of deep linear state space models. The authors find that when parameter norms are unconstrained, increasing depth and width are generally equivalent. When parameter norms are constrained, depth and width effects differ, which the authors present. Increasing depth increases capacity to represent targets with large norms with smaller-norm weights. The authors provide theoretical results as well as empirical evidence to support their findings.

**Strengths:**

The authors provide a compelling argument in the introduction to motivate the analysis of depth for expressive capacity of deep linear SSMs; and provide good foundational background on related works.

The authors provide a solid theoretical analysis (Appendix A)

**Weaknesses:**

Inconsistent related works paragraph headers.

The theoretical framework could be better integrated into the overall narrative (and some of it moved into the Appendix), currently the theory and derived results are quite dense and could distract from the key results.

Section 5 is very verbose, and can be tied together more concisely. Currently, the overwhelming amount of text hides the results.
The empirical results can be expanded to cover more complex tasks and models.

**Questions:**

N/A

---

> ### Author Response · Authors · 2025-11-22
> **Reply to weakness (part 1)**
>
> **W1: Inconsistent related works paragraph headers.**
>
> Thanks for your suggestions. We will standardize the section title in the revised version.
>
> **W2: The theoretical framework could be better integrated into the overall narrative (and some of it moved into the Appendix), currently the theory and derived results are quite dense and could distract from the key results.**
>
> We believe that the amount of details we provided for both the theoretical and the experimental part is appropriate for the readership of ICLR. If you have any specific suggestions on which part is too dense and needs to be moved to the appendix, please let us know and we can discuss. Overall, we plan to improve the intuition behind how our theory integrates with practice. In particular, we add a new experiment explaining why parameter norm control is important.
>
> We train linear SSMs with different depth and width but we  keep the same parameter counts to learn the impulse memory functions defined as equation (15) at positions $t=10$ and $t=50$ and the results are shown in the table below.
>
> | Architecture        | steps (190s), $t=10$ | loss, $t=10$                               | steps (190s), $t=50$ | loss, $t=50$                    |
> |---------------------|----------------------|----------------------------------------------------|----------------------|---------------------------------|
> | 1 layer, width 31   | 3328                 | **$2.1\times 10^{-5}$**                    | 3328                 | $1.3\times 10^{-3}$             |
> | 2 layers, width 16  | 2195                 | $2.3\times 10^{-5}$                         | 2199                 | $5.6\times 10^{-4}$             |
> | 3 layers, width 11  | 1603                 | $3.4\times 10^{-5}$                          | 1613                 | **$5.3\times 10^{-4}$**         |
>
> The impulse memory function with $t=10$ is a ''small-norm target'' whereas that with $t=50$ is a relatively ''large-norm target''.
> By this we mean that to learn the target for $t=50$ at fixed model sizes, it is known that one requires larger norms in the trainable parameters to reach a similar approximation error ([1]). All models are trained for approximately the same wall-clock time ($\approx 190$s), so deeper models see fewer optimization steps. For the impulse memory function target at $t=10$, the shallow 1-layer SSM converges fastest and attains the lowest loss. For the impulse memory function target at $t=50$ which requires larger parameter norms to reach a similar approximation error, the deepest 3-layer SSM achieves the best loss, even though it runs slower per step.
> This is consistent with our Theorem 2: depth could help reduce the norm. A deep SSM can realize a large effective kernel norm by factorizing it into a product of smaller per-layer norms, whereas a shallowm SSM cannot, making the large-norm target harder to approximate under norm constraints.
>
> **W3: Section 5 is very verbose, and can be tied together more concisely. Currently, the overwhelming amount of text hides the results. The empirical results can be expanded to cover more complex tasks and models.**
>
> We believe that the amount of details in section 5 is appropriate. We summarize our experimental setup and main findings as follows: Each of the two experiments is carefully designed to verify specific theoretical results presented in the paper. Experiments in Section 5.1 empirically confirm that increasing the number of layers leads to a decrease in the norm of weights, as stated in Theorem 4.2. The experiments in Section 5.2 extends our investigation to a nonlinear S4D model on MNIST and CIFAR-10 datasets. While our theoretical results are derived for linear models, this experiment provides qualitative evidence that increasing depth while keeping the same parameter counts still improves performance, suggesting that our theoretical insights may generalize qualitatively to nonlinear settings.  Hence, we believe the complexity of the experiments are appropriate to illustrate our theoretical insights. If you have any specific suggestions on the expansion of our experiments, please comment below.
>
> **Reference:**
>
> [1] Fusheng Liu and Qianxiao Li. Autocorrelation matters: Understanding the role of initialization
> schemes for state space models. In The Thirteenth International Conference on Learning Representations, 2025.

---

### Author Response · Authors · 2025-11-25
**Revised Manuscript Uploaded**

We would like to thank the reviewers for their valuable comments and suggestions. We have carefully revised the manuscript accordingly and have uploaded the updated version.

---

> ### Author Response · Authors · 2025-11-30
> **Summarize the Main Revisions**
>
> In the revised manuscript, we primarily focus on improving clarity, tightening the narrative around our theory and better connecting the theoretical results to experiments and practice.
>
> **1. Presentation and notation**
>
> We have corrected several citation and wording issues and now explicitly state the zero-state initialization throughout the theoretical sections. We rewrote the statements for Theorem 1, Theorem 2 and Lemma 2 to be more precise, explicitly defining notions such as the output coefficients and output expansion. We also state more clearly that Theorem 2 and Corollary 1 concern an $l$-layer SSM that is equivalent to a given one-layer SSM. This directly addresses Reviewer CCUp’s concerns about vagueness and rigor.
>
> **2.Integration of theory and experiments**
>
> Section 5 has been streamlined to make the main messages more prominent: each experiment is now explicitly tied to a specific theorem and we reduced overly verbose descriptions. Besides, for the "teacher–student'' experiment, we clarify that this is a deterministic reconstruction experiment using the constructive mapping of Theorem 2 (no SGD) and explain that the teacher and student kernels are exactly equivalent. This responds to the questions and confusion from Reviewers CCUp and XwKy. In the S4D experiment (Section 5.2), we now describe the setup more carefully. The SSM backbone is a stack of linear SSM layers with a fixed FFN, so the number of nonlinear layers is held constant while we vary depth/width under a fixed SSM parameter budget. We emphasize that this is intended as qualitative evidence that the norm‑reduction phenomenon extends to nonlinear SSMs, addressing Reviewer 8W94 and XwKy’s concerns about purpose and interpretation.
>
> **3. New experiment reporting on norms and resource trade‑offs**
>
> We have added a new experiment on learning impulse memory functions at two positions (a “small‑norm” and a “large‑norm” target) under a fixed parameter budget, comparing 1-, 2-, and 3-layer linear SSMs. We show that, for the large‑norm target, deeper SSMs achieve better loss despite having fewer optimization steps, illustrating why parameter-norm control matters and directly supporting Theorem 2. This responds to Reviewer 8W94’s questions about the significance of norm constraints.
>
> **4.Significance and practical implications**
>
> We expanded the discussion of how our results inform architecture design: (i) when norms are effectively unconstrained, depth and width are expressively equivalent and there is no expressivity-based reason to prefer deep linear SSMs; (ii) under explicit or implicit norm constraints (stability, weight decay, small initialization), deeper SSMs can realize larger effective kernel norms at the same parameter counts, providing a concrete rationale for using depth; (iii) this creates a practical trade‑off between norm control and computational cost, offering guidance on when to prefer shallow vs. deep SSMs. This responds to Reviewer 8W94’s questions on the practical implications.
>
> Overall, we would like to thank the reviewers for their valuable comments and suggestions and we have uploaded the updated version. We hope that our revisions and clarifications adequately address the reviewers’ concerns and we would be grateful if the AC could consider our paper in light of these revisions.

---

### Meta-Review · Area_Chair_WcWi · 2026-01-08

**Summary:**

The paper presents several results on linear SSMs. Specifically that without norm constraints width does not help and that with norm constraints shallow models can be represented by deeper models with lower norms.
There were diverging opinions in the review, mostly around the practical relevance of the work and how surprising the results are.
The AC has also read the paper and reviews. Few points that come up are:
1. The linear SSM is essentially a linear dynamical system that is time varying and has a delay (because hl(t+1) is connected to h_{l-1}(t+1) but also to hl(t)). Such models are standard in linear dynamical systems (LDS), and at least some of the results in the paper (e.g., impulse response) are directly inferable from this view. The authors were expected to make this connection to this extensively studied setup. It is likely that expressivity results would also come out of this.
2. Both the expressivity result and the norm result are not that surprising. The fact that a linear network does not gain from depth seems natural, though it's good to show it formally (again, I'd expect to couch this in LDS language). For norm, again it is not very surprising that depth (which is intuitively similar to multiplication) allows you to reduce the norm of each parameter separately. The authors are also encouraged to discuss whether such results are already true for regular linear networks.  The authors may also want to discuss the relation between norm constraints on parameters and rank or trace-norm minimization (see part 2 of lemma 1 in https://people.csail.mit.edu/jrennie/papers/icml05-mmmf.pdf). Also see https://proceedings.mlr.press/v80/arora18a.html for how depth can help optimization.
Given this and other issues raised by reviewers, the authors are encouraged to improve the manuscript.

**Reviewer Concerns:**

There are diverging views here, with one reviewer seeing this as too theoretical, two others quite supporting, and one giving a more superficial review.
I don't think any would change their opinion given the rebuttal (which they did read and address).

**Reviewer Scores:**

I don't think reviews would have changed, so I also read and gave a detailed meta review.

---

### Decision · Program_Chairs · 2026-01-26

Reject